# Rethinking the Role of Token Retrieval in Multi-Vector Retrieval

**Jinhyuk Lee**[*]   **Zhuyun Dai**   **Sai Meher Karthik Duddu**

**Tao Lei**   **Iftekhar Naim**   **Ming-Wei Chang**   **Vincent Y. Zhao**

Google DeepMind

## Abstract

Multi-vector retrieval models such as ColBERT [Khattab and Zaharia, 2020] allow token-level interactions between queries and documents, and hence achieve state of the art on many information retrieval benchmarks. However, their non-linear scoring function cannot be scaled to millions of documents, necessitating a three-stage process for inference: retrieving initial candidates via token retrieval, accessing all token vectors, and scoring the initial candidate documents. The non-linear scoring function is applied over all token vectors of each candidate document, making the inference process complicated and slow. In this paper, we aim to simplify the multi-vector retrieval by rethinking the role of token retrieval. We present XTR, Conte**X**tualized **T**oken **R**etriever, which introduces a simple, yet novel, objective function that encourages the model to retrieve the most important document tokens first. The improvement to token retrieval allows XTR to rank candidates only using the retrieved tokens rather than all tokens in the document, and enables a newly designed scoring stage that is two-to-three orders of magnitude cheaper than that of ColBERT. On the popular BEIR benchmark, XTR advances the state-of-the-art by 2.8 nDCG@10 without any distillation. Detailed analysis confirms our decision to revisit the token retrieval stage, as XTR demonstrates much better recall of the token retrieval stage compared to ColBERT.

## 1   Introduction

The performance of a dense retrieval model is largely affected by how it defines expressive representations over queries and documents, and whether it can efficiently retrieve and score a document using these vector representations. For example, dual encoder models [Yih et al., 2011, Lee et al., 2019, Karpukhin et al., 2020, Ni et al., 2021] encode queries and documents into single vectors and compute query-document similarities using dot products. While these models are very efficient for retrieval, their expressivity is limited due to the absence of token-level modeling for scoring. In contrast, multi-vector models such as ColBERT [Khattab and Zaharia, 2020, Santhanam et al., 2022b] are directly designed to capture token-level interactions. By utilizing a (non-linear) scoring function over all query and document token representations, multi-vector models enjoy much better model expressivity and often achieve superior results across various benchmarks [Thakur et al., 2021].

The enhanced model expressivity, however, comes at a great cost of inference complexity. Unlike the case in dual encoders, the non-linear scoring function in multi-vector retrieval models prohibits the use of efficient Maximum Inner Product Search (MIPS) [Ram and Gray, 2012, Shrivastava and Li, 2014, 2015, Shen et al., 2015] for finding the maximum scoring documents. As a result, models such as ColBERT adopt an intricate and resource-intensive inference pipeline, which typically consists

---

[*]Correspondence: `jinhyuklee@google.com`

37th Conference on Neural Information Processing Systems (NeurIPS 2023).

of three stages: 1) *token retrieval:* using each query token to retrieve document tokens, with their source documents becoming candidates; 2) *gathering:* collecting all the token embeddings from each candidate document, including those that are not retrieved in the first stage (most document tokens are not retrieved); and 3) *scoring:* ranking candidates using a non-linear function based on all the token embeddings per document.

This procedure leads to two major issues. First, compared to the token retrieval stage, gathering all document token embeddings and re-scoring the documents can introduce orders of magnitude additional data loading and floating operation cost, making multi-vector models extremely expensive to deploy. Secondly, while the candidate documents are decided in the token retrieval stage, previous training objectives are designed for the scoring stage. This creates a significant training-inference gap causing multi-vector models achieve sub-optimal (and often poor) recall performance. Clearly, the three-stage pipeline has largely limited the potential of multi-vector models, raising an interesting research question – *can the token retrieval stage alone be sufficient for great performance?*

We present XTR, Cont**X**extualized **T**oken **R**etriever: a simplified and efficient method for multi-vector retrieval, through re-thinking the role of token retrieval. The key insight of XTR is that the token retrieval in multi-vector models should be trained to retrieve the most salient and informative document tokens, so that the score between a query and document can be computed using only the retrieved information, just like how single-vector retrieval models work. By doing so, the gathering step can be completely eliminated, and the cost of scoring is significantly reduced as only a fraction of the tokens need to be considered and the dot products from the token retrieval can be reused. To improve the quality of the token retrieval, XTR proposes a novel, yet simple, training objective, which dramatically improves retrieval accuracy, doubling the chances of a gold token being retrieved in the top-$k$ results. Furthermore, despite the improved token retrieval, some relevant tokens may still be missed (i.e., not retrieved). To address this issue, we propose a simple method, called missing similarity imputation, which accounts for the contribution of the missing tokens to the overall score.

XTR streamlines the inference process, bringing it closer to the straightforward procedure of dual encoders, while maintaining and enhancing the expressive scoring function of multi-vector retrieval models. On the BEIR [Thakur et al., 2021] and LoTTE [Santhanam et al., 2022b] benchmarks, XTR attains state-of-the-art performance, requiring neither distillation nor hard negatiave mining. Notably, our model surpasses state-of-the-art dual-encoder GTR [Ni et al., 2021] by 3.6 nDCG@10 on BEIR without any additional training data. On the EntityQuestions benchmark [Sciavolino et al., 2021], XTR outperforms the previous state-of-the-art by 4.1 points on top-20 retrieval accuracy. XTR also does not require any secondary pre-training for retrieval and greatly outperforms mContriever [Izacard et al., 2022] on MIRACL, which contains multilingual retrieval tasks in 18 languages [Zhang et al., 2022b]. Our analysis supports that XTR indeed benefits from retrieving more contextualized tokens in relevant contexts, while making the scoring stage two-to-three orders of magnitude cheaper.

## 2 Background

### 2.1 Multi-vector Retrieval

Single-vector retrieval models, also known as dual encoders, encode an input text sequence as a single dense embedding and define the similarity of a query and a document based on the dot product [Lee et al., 2019, Karpukhin et al., 2020]. Multi-vector retrieval models, on the other hand, make use of multiple dense embeddings for each query and document, typically leveraging all contextualized word representations of the input to gain improved model expressivity.

Consider a query $Q = \{\mathbf{q}_i\}_{i=1}^n$ and a document $D = \{\mathbf{d}_j\}_{j=1}^m$ where $\mathbf{q}_i$ and $\mathbf{d}_j$ denote the $d$-dimensional query token vector and the document token vector, respectively. Multi-vector retrieval models compute the query-document similarity as follows: $f(Q, D) = \sum_{i=1}^n \sum_{j=1}^m \mathbf{A}_{ij} \mathbf{P}_{ij}$ where $\mathbf{P}_{ij} = \mathbf{q}_i^\top \mathbf{d}_j$ and $\mathbf{A} \in \{0, 1\}^{n \times m}$ denotes the alignment matrix with $\mathbf{A}_{ij}$ being the token-level alignment between the query token vector $\mathbf{q}_i$ and the document token vector $\mathbf{d}_j$. The sum-of-max operator of ColBERT [Khattab and Zaharia, 2020] sets $\mathbf{A}_{ij} = \mathbb{1}_{[j=\mathrm{argmax}_{j'}(\mathbf{P}_{ij'})]}$ where the argmax operator is over $1 \le j' \le m$ (i.e., tokens from a single document $D$) and $\mathbb{1}_{[*]}$ is an indicator function. Then, $f_{\mathrm{ColBERT}}(Q, D)$ is defined as follows:

$$f_{\mathrm{ColBERT}}(Q, D) = \frac{1}{n} \sum_{i=1}^n \sum_{j=1}^m \mathbf{A}_{ij} \mathbf{P}_{ij} = \frac{1}{n} \sum_{i=1}^n \max_{1 \le j \le m} \mathbf{q}_i^\top \mathbf{d}_j. \tag{1}$$

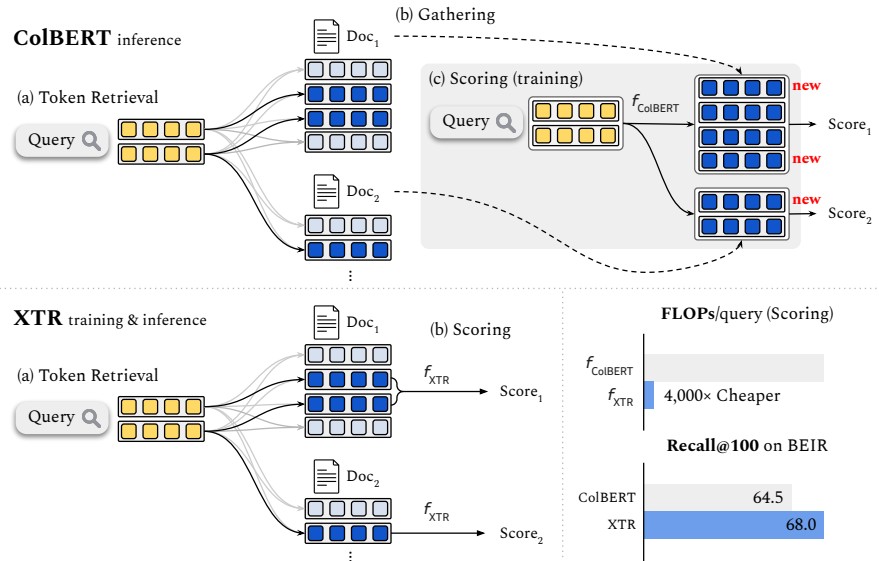

Figure 1: Overview of XTR. ColBERT has the three-stage inference combining (a) the token retrieval, (b) the gathering and (c) the scoring stages (§2.2). XTR leverages the token retrieval for both training and inference. XTR efficiently obtains the score of each candidate document by applying $f_{\text{XTR}}$ (or $f_{\text{XTR}'}$) on the retrieved tokens, completely removing the gathering stage (§3.2).

Here, we include the normalizer $n$, which was not included in the original sum-of-max, as it stabilizes training while not affecting the ranking during inference. After computing the query-document similarity, multi-vector retrieval models are typically trained with a cross-entropy loss over in-batch negatives [Santhanam et al., 2022b, Qian et al., 2022]. Specifically, given a positive document $D^+$ for $Q$ and a set of mini-batch documents $D_{1:B} = [D_1, \ldots, D_B]$ where $D^+ \in D_{1:B}$, they minimize the cross-entropy loss defined as: $\mathcal{L}_{\text{CE}} = -\log \frac{\exp f(Q, D^+)}{\sum_{b=1}^{B} \exp f(Q, D_b)}$.

## 2.2 Three-stage inference of Multi-vector Retrieval

Unlike dual encoder models, finding the maximum scoring document—the document that maximizes eq. (1)—cannot be directly handled by MIPS as the scoring function uses a non-linear, sum-of-max operation. Instead, a multi-vector retrieval model typically takes the following steps for the inference. 1) **Token Retrieval**: for each of the $n$ query token vectors, it first retrieves $k'$ document token vectors, which is simply used to form *initial candidate document set* by taking the union of source documents of retrieved tokens. The total number of candidate documents is up to $nk'$ if each token is coming from a unique document.[2] 2) **Gathering**: since the scoring function eq. (1) requires the computation over all document tokens, multi-vector models need to load all of the token vectors of the candidate documents. To optimize the loading process, a RAM-based index is often employed. 3) **Scoring**: to provide final ranks of candidate documents, multi-vector models score all the candidate documents with eq. (1). This stage is also called *refinement*. Note that the training of typical multi-vector models only takes care of the scoring stage with mini-batch documents. Finally, top-$k$ documents are returned based on the computed scores. The three-stage inference is illustrated in the top of Figure 1.

# 3 XTR: Contextualized Token Retriever

Unlike existing multi-vector models that follow the retrieve-gather-score stages, XTR directly scores documents utilizing the tokens retrieved from the token retrieval stage. In this section, we start by showing why the existing cross entropy loss with the sum-of-max scoring function would fail on the first-stage token retrieval. Then, we introduce simple but important modifications for XTR.

---

[2]In fact, each candidate document of a T5-based ColBERT is retrieved by 1.48 tokens per on average, meaning that the most of the candidate documents are unique.

Given a positive document $D^+$ and a set of negative documents $D^-_{1:r} = [D^-_1, \ldots, D^-_r]$ for a query $Q$, the first-stage token retrieval needs to retrieve the tokens of $D^+$, but not the tokens of negative documents. However, the following example shows that the sum-of-max operator used by ColBERT is not specifically designed to retrieve tokens of relevant documents.

**Failure case** Assume that $f_{\text{ColBERT}}(Q, D^+) = 0.8$ where all the individual max token similarity (i.e., $\mathbf{q}_i^\top \mathbf{d}_j^+$ where $\mathbf{A}_{ij} = 1$) is 0.8. On the other hand, assume $f_{\text{ColBERT}}(Q, D^-) = 0.2$ for all $D^- \in D^-_{1:r}$ where each $D^-$ has a highly peaked token similarity greater than 0.8 but others close to zero (i.e., there exists $\mathbf{q}_i^\top \mathbf{d}_j^- > 0.8$ where $\mathbf{A}_{ij} = 1$ while other $\mathbf{q}_i^\top \mathbf{d}_j^- \to 0$). Since the sum-of-max operator only cares about the document-level scores, the cross entropy loss would be close to zero during training.[3]

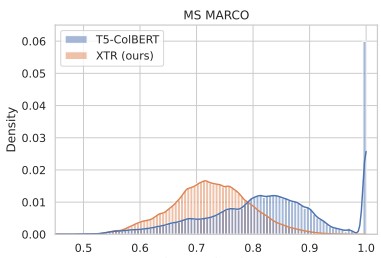

Figure 2: Density histogram of 4,000 token retrieval scores (cosine similarity). Training with $f_{\text{ColBERT}}$ (T5-ColBERT; §4) causes many document tokens to have extremely high scores regardless of their actual relevance with respect to the input query tokens. XTR mitigates this problem with a better training objective.

However, for each of $n$ query tokens, if there exists at least one negative document token that has a high token similarity greater than 0.8, the token retrieval with top-$k' = 1$ would fail to retrieve any tokens of $D^+$. As a result, multi-vector retrieval model with the sum-of-max operator will not be able to lower the high scores of some negative tokens. Figure 2 shows that the sum-of-max training causes many document tokens to have unreasonably high scores regardless of their actual relevance to the query tokens.

## 3.1 In-Batch Token Retrieval

To train multi-vector retrieval models to directly retrieve tokens of relevant documents, we simulate the token retrieval stage during training. This can be simply achieved by employing a different alignment strategy $\hat{\mathbf{A}}$. Specifically, we set the alignment $\hat{\mathbf{A}}_{ij} = \mathbb{1}_{[j \in \text{top-}k_{j'}(\mathbf{P}_{ij'})]}$ where the top-$k$ operator is applied over $1 \leq j' \leq mB$ (i.e., tokens from $B$ mini-batch documents) returning the indices of $k$ largest values. During training, we use a hyperparameter $k_{\text{train}}$ for the top-$k$ operator. Then, we simply modify eq. (1) as follows:

$$f_{\text{XTR}}(Q, D) = \frac{1}{Z} \sum_{i=1}^{n} \max_{1 \leq j \leq m} \hat{\mathbf{A}}_{ij} \mathbf{q}_i^\top \mathbf{d}_j. \tag{2}$$

The intuition is that we consider the token similarities within $D$ only when they are high enough to be retrieved within top-$k_{\text{train}}$ from a mini-batch. Here, we use a normalizer $Z = |\{i | \exists j, s.t. \hat{\mathbf{A}}_{ij} > 0\}|$, which is essentially the number of query tokens that retrieved at least one document token of $D$.[4] If all $\hat{\mathbf{A}}_{ij} = 0$, we clip $Z$ to a small number and $f_{\text{XTR}}(Q, D)$ becomes 0. As a result, our model cannot assign a high token similarity to negative documents as it blocks tokens of positive documents to be retrieved. With the previous failure case where $f_{\text{ColBERT}}$ assigned a high score on $D^+$ even though it cannot be retrieved, our similarity function incurs a high loss as $f_{\text{XTR}}(Q, D^+) = 0$ during training (since tokens of $D^+$ were not retrieved). For training, we use the same cross entropy loss defined in §2.1 with our new scoring function. Note that the training data only contains document-level annotations, but XTR encourages important tokens from positive documents to be retrieved.

## 3.2 Scoring Documents using Retrieved Tokens

During inference, multi-vector retrieval models first have a set of candidate documents $\hat{D}_{1:C}$ from the token retrieval stage:

$$\hat{D}_{1:C} = \{\hat{D} | d_j \in \hat{D} \wedge d_j \in \text{top-}k'(\mathbf{q}_*)\}. \tag{3}$$

---

[3]Indeed, our derivative analysis in Appendix A shows that the token-level similarity would not change if the document-level scores are already well discriminated.

[4]We tried different normalizers such as $n$ and found that $Z$ works the best while stabilizing the training.

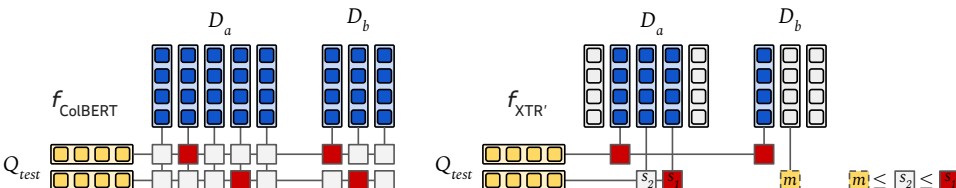

Figure 3: Comparison of $f_{\text{ColBERT}}$ in eq. (1) and $f_{\text{XTR}'}$ in eq. (4). Assume that $D_a$ and $D_b$ were selected as initial candidate documents from the token retrieval stage. $f_{\text{ColBERT}}$ loads all token vectors of $D_a$ and $D_b$ and exhaustively recomputes pairwise token similarity to obtain the max values (**red** boxes). On the other hand, $f_{\text{XTR}'}$ does not load any token vectors and reuses retrieval scores from the first-stage token retrieval. Assume that, with the top-2 token retrieval results, the first query token retrieved each max score of $D_a$ and $D_b$, but the second query token retrieved two tokens only from $D_a$ but not $D_b$. We impute the missing similarity $m$ for $D_b$ (denoted as **yellow** dashed box) by finding its upper bound using the top-2 score (denoted as $s_2$) of the second query token (i.e., $m \le s_2 \le s_1$).

| | Scoring | Estimated FLOPs/query | Setting |
|---|---|---|---|
| $f_{\text{ColBERT}}$ | $n^2 k'(2\bar{m}d + \bar{m} + 1)$ | $0.36 \times 10^9$ | $M = 3 \times 10^9, n = 16, d = 128,$ |
| $f_{\text{XTR}'}$ | $n^2 k'(\bar{r} + 1)$ | $0.09 \times 10^6$ | $k' = 100, \bar{m} = 55, \bar{r} = 2.5$ |

Table 1: FLOPs comparison of ColBERT and XTR for the scoring stage. XTR only adds minimal complexity for scoring each candidate document. The setting is derived from MS MARCO.

Here, top-$k'(\mathbf{q}_*)$ is a union of top-$k'$ document tokens (from the entire corpus) based on the inner product scores with each query vector (i.e., $\mathbf{q}^\top \mathbf{d}$). Given the $n$ query token vectors, there are $C$ ($\le nk'$) candidate documents. Previous methods load the entire token vectors of each document and compute eq. (1) for every query and candidate document pair, which takes $\mathcal{O}(n^2 k' \bar{m} d)$ computation per query ($\bar{m}$ = average document length). Instead, we propose to score the documents *solely using the retrieved token similarity*. This significantly reduces the computational cost for the scoring stage since re-using the token retrieval scores removes computing redundant inner products and unnecessary (non-max) inner products. Furthermore, the expensive gathering stage (which requires loading all the document token vectors for computing eq. (1)) can be removed completely. Unlike previous work [Macdonald and Tonellotto, 2021] that leverages token retrieval to sort first-stage candidate documents before the scoring stage, we aim to directly provide the final scores of documents.

**Missing similarity imputation** During inference, we retrieve $k'$ document tokens for each of $n$ query tokens. Assume that each document token belongs to a unique document, providing $C = nk'$ candidate documents in total. This leaves us with a single token similarity to score each document in the absence of the gathering stage. However, during training—either with eq. (1) or eq. (2)—each positive document has up to $n$ (max) token similarities to average, which mostly converges to $n$ as training proceeds. Hence, during inference, we impute the *missing similarity* for each query token treating each of candidate documents as if it were positive with $n$ token similarities.

For every candidate document $\hat{D}$, we first define the following scoring function for the inference:

$$f_{\text{XTR}'}(Q, \hat{D}) = \frac{1}{n} \sum_{i=1}^{n} \max_{1 \le j \le m} \left[ \hat{\mathbf{A}}_{ij} \mathbf{q}_i^\top \mathbf{d}_j + (1 - \hat{\mathbf{A}}_{ij}) m_i \right]. \tag{4}$$

This is similar to eq. (2), but introduces $m_i \in \mathbb{R}$, which estimates the missing similarity for each $q_i$. $\hat{\mathbf{A}}$ is defined similar to the one described in eq. (2) except that it uses $k'$ for the top-$k$ operator. Each $q_i$ would take the missing similarity $m_i$ as the maximum value if $\hat{\mathbf{A}}_{i*} = 0$ and $m_i \ge 0$. Importantly, $f_{\text{XTR}'}$ removes the need of recomputing any $\mathbf{q}_i^\top \mathbf{d}_j$ since when $\hat{\mathbf{A}}_{ij} = 1$ we already know the retrieval score from the token retrieval stage, and when $\hat{\mathbf{A}}_{ij} = 0$ we simply don't need to compute it as $\hat{\mathbf{A}}_{ij} \mathbf{q}_i^\top \mathbf{d}_j = 0$. Note that when every $\hat{\mathbf{A}}_{ij} = 1$, the equation becomes the sum-of-max operator. On the other hand, when no document tokens of $\hat{D}$ were retrieved for $q_i$ (i.e., $\hat{\mathbf{A}}_{i*} = 0$), we fall back to the imputed score $m_i$, which provides an approximated sum-of-max result.

| | MS | AR | TO | FE | CF | SF | CV | NF | NQ | HQ | FQ | SD | DB | QU | Avg. |
|---|---|---|---|---|---|---|---|---|---|---|---|---|---|---|---|
| *One Retriever per Domain* | | | | | | | | | | | | | | | |
| GenQ | 40.8 | 49.3 | 18.2 | 66.9 | 17.5 | 64.4 | 61.9 | 31.9 | 35.8 | 53.4 | 30.8 | 14.3 | 32.8 | 83.0 | 43.1 |
| PTR$_{\text{retriever}}$ | - | 58.8 | 25.6 | 76.2 | 23.5 | 63.8 | 70.2 | 33.7 | 45.6 | 61.7 | 43.0 | 18.3 | 34.4 | 87.5 | 49.4 |
| *One Retriever for All* | | | | | | | | | | | | | | | |
| BM25 | 22.8 | 31.5 | 36.7 | 75.3 | 21.3 | 66.5 | 65.6 | 32.5 | 32.9 | 60.3 | 23.6 | 15.8 | 31.3 | 78.9 | 44.0 |
| ColBERT | 40.1 | 23.3 | 20.2 | 77.1 | 18.4 | 67.1 | 67.7 | 30.5 | 52.4 | 59.3 | 31.7 | 14.5 | 39.2 | 85.4 | 45.1 |
| GTR$_{\text{base}}$ | 42.0 | 51.1 | 21.5 | 66.0 | 24.1 | 60.0 | 53.9 | 30.8 | 49.5 | 53.5 | 34.9 | 14.9 | 39.2 | 88.1 | 45.2 |
| T5-ColBERT$_{\text{base}}$ | 45.6 | 28.8 | 31.1 | 72.4 | 18.1 | 70.4 | 68.3 | 34.0 | 52.2 | 61.7 | 33.4 | 14.1 | 41.6 | 82.3 | 46.8 |
| **XTR$_{\text{base}}$** | 45.0 | 40.7 | 31.3 | 73.7 | 20.7 | 71.0 | 73.6 | 34.0 | 53.0 | 64.7 | 34.7 | 14.5 | 40.9 | 86.1 | 49.1 |
| Splade$_{\text{v2}}$ ♣♦ | 43.3 | 47.9 | 27.2 | 78.6 | 23.5 | 69.3 | 71.0 | 33.4 | 52.1 | 68.4 | 33.6 | 15.8 | 43.5 | 83.8 | 49.9 |
| ColBERT$_{\text{v2}}$ ♣♦ | - | 46.3 | 26.3 | 78.5 | 17.6 | 69.3 | 73.8 | 33.8 | 56.2 | 66.7 | 35.6 | 15.4 | 44.6 | 85.2 | 49.9 |
| GTR$_{\text{xxl}}$ | 44.2 | 54.0 | 23.3 | 74.0 | 26.7 | 66.2 | 50.1 | 34.2 | 56.8 | 59.9 | 46.7 | 16.1 | 40.8 | 89.2 | 49.1 |
| T5-ColBERT$_{\text{xxl}}$ | 47.3 | 33.8 | 31.0 | 74.2 | 19.7 | 73.1 | 75.8 | 35.2 | 60.5 | 65.2 | 43.5 | 17.1 | 45.0 | 86.0 | 50.8 |
| **XTR$_{\text{xxl}}$** | 46.6 | 44.2 | 30.9 | 77.0 | 24.5 | 74.3 | 78.9 | 35.3 | 60.9 | 66.2 | 43.8 | 17.1 | 44.3 | 88.1 | **52.7** |

| | LoTTE Search | | | | | | LoTTE Forum | | | | | |
|---|---|---|---|---|---|---|---|---|---|---|---|---|
| | Writing | Rec. | Sci. | Tech. | Life. | Pooled | Writing | Rec. | Sci. | Tech. | Life. | Pooled |
| BM25 | 60.3 | 56.5 | 32.7 | 41.8 | 63.8 | 48.3 | 64.0 | 55.4 | 37.1 | 39.4 | 60.6 | 47.2 |
| ColBERT | 74.7 | 68.5 | 53.6 | 61.9 | 80.2 | 67.3 | 71.0 | 65.6 | 41.8 | 48.5 | 73.0 | 58.2 |
| GTR$_{\text{base}}$ | 74.1 | 65.7 | 49.8 | 58.1 | 82.0 | 65.0 | 69.2 | 62.0 | 33.7 | 47.6 | 72.2 | 54.9 |
| **XTR$_{\text{base}}$** | 77.0 | 69.4 | 54.9 | 63.2 | 82.1 | 69.0 | 73.9 | 68.7 | 42.2 | 51.9 | 74.4 | 60.1 |
| Splade$_{\text{v2}}$ ♣♦ | 77.1 | 69.0 | 55.4 | 62.4 | 82.3 | 68.9 | 73.0 | 67.1 | 43.7 | 50.8 | 74.0 | 60.1 |
| ColBERT$_{\text{v2}}$ ♣♦ | 80.1 | 72.3 | 56.7 | 66.1 | 84.7 | 71.6 | 76.3 | 70.8 | 46.1 | 53.6 | 76.9 | 63.4 |
| GTR$_{\text{xxl}}$ | **83.9** | 78.0 | 60.0 | 69.5 | 87.4 | 76.0 | 79.5 | 73.5 | 43.1 | 62.6 | 81.9 | 66.9 |
| **XTR$_{\text{xxl}}$** | 83.3 | **79.3** | **60.8** | **73.7** | **89.1** | **77.3** | **83.4** | **78.4** | **51.8** | **64.5** | **83.9** | **71.2** |

♣: cross-encoder distillation ♦: model-based hard negatives

Table 2: (top) nDCG@10 on MS MARCO (in-domain) and BEIR (zero-shot). The last column shows the average over 13 BEIR datasets. (bottom) Top-5 retrieval accuracy on LoTTE datasets (zero-shot).

In fact, we can find the upper bound of the missing similarity. For every token retrieval with $\mathbf{q}_i$, the missing similarity of the query token for $\hat{D}$ will be upper bounded by its last top-$k'$ score. Specifically, for each query token $q_i$, we have the following top-$k'$ token similarity during inference: $[\mathbf{q}_i^\top \mathbf{d}_{(1)}, \ldots \mathbf{q}_i^\top \mathbf{d}_{(k')}]$. Here, each $\mathbf{d}_{(*)}$ could come from a different document. Since the missing similarity would have a score less than equal to the score of the last retrieved token, we know that $m_i \leq \mathbf{q}_i^\top \mathbf{d}_{(k')}$. With a larger $k'$, the upper bound becomes tighter. In our experiments, we show that simply choosing $m_i = \mathbf{q}_i^\top \mathbf{d}_{(k')}$ works well especially when a model is trained with $f_{\text{XTR}}$.[5] While we also tried more complicated imputation methods based on regression, our method was competitive enough despite its simplicity. The imputation process is illustrated in Figure 3.

Table 1 shows the estimated FLOPs of ColBERT and XTR (see Appendix B for more details). Due to the differences in hardware and infrastructure, we mainly compared the theoretical FLOPs. XTR reduces the FLOPs at the scoring stage by 4000× making multi-vector retrieval more efficient.

## 4 Experiments

**Experimental Setting** Following Ni et al. [2021], we fine-tune XTR on MS MARCO with a fixed set of hard negatives from RocketQA [Qu et al., 2021]. Then, we test XTR on MS MARCO (MS; in-domain) and zero-shot IR datasets. For the zero-shot evaluation, we use 13 datasets from BEIR [Thakur et al., 2021] (see Appendix C for acronyms), 12 datasets from LoTTE [Santhanam et al., 2022b], and 4 datasets on open-domain QA passage retrieval (EQ: EntityQuestions [Sciavolino et al., 2021], NQ, TQA: TriviaQA, SQD: SQuAD). We also train multilingual XTR (mXTR) and evaluate it on MIRACL [Zhang et al., 2022b], which contains retrieval tasks in 18 languages. The performance gap between T5-ColBERT [Qian et al., 2022] and XTR shows the improvement with our methods on a multi-vector retrieval model. For implementation details and baselines, see Appendix C. For the relationship between hyperparameters (e.g., $k_{\text{train}}$ and $k'$), see §5.3.

---

[5]We found that directly training with $f_{\text{XTR}'}$ instead of $f_{\text{XTR}}$ fails to converge, which we leave as future work.

| | EQ | | NQ | | TQA | | SQD | |
|---|---|---|---|---|---|---|---|---|
| | Top-20 | Top-100 | Top-20 | Top-100 | Top-20 | Top-100 | Top-20 | Top-100 |
| BM25♣ | 71.4 | 80.0 | 62.9 | 78.3 | 76.4 | 83.2 | 71.1 | 81.8 |
| DPR$_{multi}$ + BM25♣ | 73.3 | 82.6 | 82.6 | 88.6 | 82.6 | 86.5 | 75.1 | 84.4 |
| ART$_{MS\ MARCO}$♦ | 75.3 | 81.9 | - | - | 78.0 | 84.1 | 68.4 | 80.4 |
| GTR$_{base}$♦ | 73.3 | 80.6 | 78.5 | 86.5 | 76.2 | 83.4 | 65.9 | 77.6 |
| GTR$_{xxl}$♦ | 75.3 | 82.5 | 83.5 | 89.8 | 81.7 | 86.6 | 70.4 | 80.6 |
| DPR$_{multi}$ | 56.7 | 70.0 | 79.5 | 86.1 | 78.9 | 84.8 | 52.0 | 67.7 |
| ColBERT | - | - | 79.1 | - | 80.3 | - | 76.5 | - |
| **XTR$_{base}$** | 79.0 | 85.2 | 79.3 | 88.1 | 80.3 | 85.5 | 78.2 | 85.9 |
| **XTR$_{xxl}$** | **79.4** | **85.9** | **84.9** | **90.5** | **83.3** | **87.1** | **81.1** | **87.6** |

♣: sparse component     ♦: retrieval pre-training

Table 3: Zero-shot passage retrieval accuracy on open-domain question answering datasets. In-domain performances are underlined and all the other performances are based on the zero-shot evaluation. For EntityQuestions, we report macro-averaged performances over different relations.

| | ar | bn | en | es | fa | fi | fr | hi | id | ja | ko | ru | sw | te | th | zh | de | yo | Avg. |
|---|---|---|---|---|---|---|---|---|---|---|---|---|---|---|---|---|---|---|---|
| BM25 | 48.1 | 50.8 | 35.1 | 31.9 | 33.3 | 55.1 | 18.3 | 45.8 | 44.9 | 36.9 | 41.9 | 33.4 | 38.3 | 49.4 | 48.4 | 18.0 | - | - | - |
| mDPR | 49.9 | 44.3 | 39.4 | 47.8 | 48.0 | 47.2 | 43.5 | 38.3 | 27.2 | 43.9 | 41.9 | 40.7 | 29.9 | 35.6 | 35.8 | 51.2 | - | - | - |
| BM25 + mDPR | 67.3 | 65.4 | 54.9 | **64.1** | **59.4** | 67.2 | 52.3 | 61.6 | 44.3 | 57.6 | 60.9 | 53.2 | 44.6 | 60.2 | 59.9 | 52.6 | - | - | - |
| *Trained on English MS MARCO* | | | | | | | | | | | | | | | | | | | |
| mContriever (en) | 55.3 | 54.2 | 37.9 | 34.1 | 42.6 | 51.2 | 31.5 | 40.6 | 36.8 | 38.3 | 46.2 | 39.9 | 44.4 | 48.7 | 52.4 | 27.4 | 32.9 | 32.9 | 41.5 |
| **mXTR$_{base}$ (en)** | 66.1 | 64.7 | 49.4 | 40.5 | 47.9 | 62.2 | 37.5 | 51.4 | 46.9 | 56.8 | 64.0 | 49.8 | 43.0 | 67.7 | 69.2 | 47.2 | 34.5 | 40.6 | 52.2 |
| **mXTR$_{xxl}$ (en)** | 74.1 | 75.5 | **56.0** | 52.4 | 56.1 | 75.1 | 51.4 | **61.8** | 52.0 | 68.7 | 67.4 | 61.3 | 69.7 | 76.0 | 76.9 | 56.9 | 51.7 | 60.3 | 63.5 |
| *Trained on English MS MARCO + MIRACL (16 languages)* | | | | | | | | | | | | | | | | | | | |
| mContriever | 64.6 | 66.4 | 41.2 | 40.3 | 46.3 | 61.9 | 42.9 | 41.9 | 44.6 | 55.6 | 55.4 | 48.1 | 65.3 | 77.6 | 69.3 | 45.9 | 39.6 | 41.9 | 52.7 |
| **mXTR$_{base}$** | 73.0 | 73.9 | 46.1 | 42.6 | 51.0 | 70.5 | 39.3 | 51.3 | 54.2 | 62.3 | 67.7 | 54.5 | 69.7 | 80.7 | 76.1 | 51.4 | 36.1 | 46.8 | 58.2 |
| **mXTR$_{xxl}$** | **77.8** | **78.4** | 52.5 | 48.9 | 56.0 | **76.0** | **52.9** | 61.5 | **54.9** | **73.4** | **68.5** | **66.2** | **79.4** | **84.3** | 80.7 | **58.9** | 52.8 | **62.4** | **65.9** |

Table 4: nDCG@10 on 18 multilingual retrieval tasks from MIRACL. Each row shows the performance of a single multilingual retrieval model. The last two surprise languages (de and yo) are not included in the training dataset of MIRACL. The last column shows the average over 18 languages.

## 4.1  In-domain Document Retrieval

**MS MARCO**   The first column of Table 2 (top) shows nDCG@10 on MS MARCO (see Table D.1 for recall@100). XTR outperforms most models and remains competitive with T5-ColBERT. This is encouraging since XTR significantly reduces the cost of the gathering–scoring stage. Note that MS MARCO may fail to reflect the actual improvement of state-of-the-art [Arabzadeh et al., 2022].

## 4.2  Zero-shot Document Retrieval

**BEIR & LoTTE**   Table 2 (top; except the first columns shows nDCG@10 on BEIR (see Table D.1 for recall@100). XTR$_{xxl}$ achieves the new state-of-the-art performances significantly outperforming both per-domain models and single model state-of-the-art. Simply scaling XTR removes the needs of designing distillation or hard negative mining pipelines [Santhanam et al., 2022b, Formal et al., 2021]. Results on LoTTE (Table 2 bottom) also show that XTR$_{base}$ is better than ColBERT and competitive with distillation-based models while XTR$_{xxl}$ advances the state-of-the-art.

**Passage retrieval for open-domain QA**   Table 3 shows results on four open-domain QA datasets. While previous work often includes sparse retrievers (e.g., BM25) [Chen et al., 2021] or contrastive pre-training [Ram et al., 2022, Sachan et al., 2022a,b] to achieve better performances on EntityQuestions, XTR simply fine-tuned on MS MARCO achieves the state-of-the-art performance.

## 4.3  Multilingual Document Retrieval

**MIRACL**   Since XTR does not need any secondary pre-training, we expect it to be better at multilingual retrieval by better utilizing the multilingual language models. We train a multilingual version of XTR with mT5 [Xue et al., 2021] and test it on multilingual retrieval tasks in 18 languages. Table 4 shows that mXTR greatly outperforms mContriever that uses expensive contrastive pre-training, as well as the hybrid model, BM25 + mDPR.

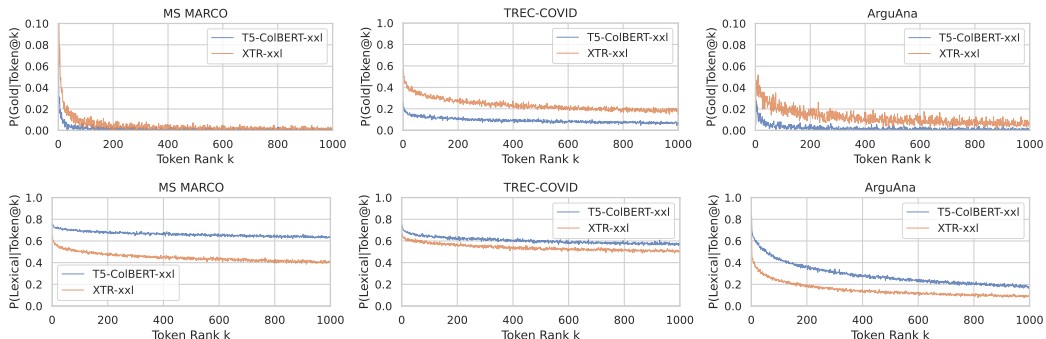

Figure 4: (top) Gold token retrieval performances of T5-ColBERT and XTR. We plot the probability of each retrieved document token at rank $k$ coming from the gold document. (bottom) Lexical token retrieval performances of T5-ColBERT and XTR. We plot the probability of each retrieved document token at rank $k$ being lexically identical to its query token.

| Model | Imputation | MRR@10 | R@1000 |
|---|---|---|---|
| T5-ColBERT$_{base}$ | None | 0.0 | 0.0 |
| | top-$k'$ score | 27.7 | 91.8 |
| XTR$_{base}$ | None | 22.6 | 88.7 |
| | $m_i = 0$ | 36.2 | 97.3 |
| | $m_i = 0.2$ | 36.4 | 97.3 |
| | top-$k'$ score | **37.4** | **98.0** |

Table 5: Impact of training objectives and imputation methods comparing T5-ColBERT and XTR. For both models, we apply $f_{XTR'}$ during inference. We report MRR@10 and Recall@1000 on the MS MARCO development set.

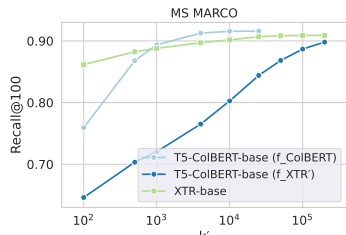

Figure 5: Recall@100 of XTR and T5-ColBERT with different $k'$. For T5-ColBERT, we use either $f_{XTR'}$ or $f_{ColBERT}$.

## 5 Analysis

### 5.1 Towards Better Token Retrieval

**Gold token retrieval**    If the tokens of gold documents are not retrieved at all, multi-vector retrieval models would fail to retrieve the gold documents. Hence, a better token retrieval would contain these *gold tokens* more often in their top results. In Figure 4 (top), we show the probability of a token at the rank $k$ coming from the gold documents of a query. To compute the probability for the rank $k$, we simply count the number of an event where a token at rank $k$ belongs to the gold document and divide it by the number of tokens at rank $k$. While this is measuring the precision of the token retrieval, we observed a similar trend for the recall of gold tokens. Compared to T5-ColBERT, XTR retrieves gold tokens with higher probability, even on MS MARCO. This shows that the training objective of XTR encourages it to retrieve tokens from more relevant context.

**Lexical token retrieval**    In Figure 4 (bottom), we show the probability of a token at the rank $k$ being the same as its query token (e.g., 'insulin' retrieving 'insulin's). T5-ColBERT has very high probability of retrieving the same token across different ranks and datasets. However, it is unclear to what extent the token retrieval stage should behave as sparse retrieval, as it might suffer from the vocabulary mismatch problem. XTR effectively lowers the reliance on the lexical matching while preserving a good amount of lexical precision so that it would achieve a high retrieval accuracy on the entity-centric dataset (§4.2). In fact, Table 6 in Appendix shows that having lower lexical matching doesn't necessarily mean a lower retrieval quality, but often means better contextualization.

### 5.2 Efficient Scoring

In Table 5, we show how we can employ the efficient scoring function $f_{XTR'}$ in XTR with minimal performance losses. We apply $f_{XTR'}$ on both T5-ColBERT and XTR, and show their performances on MS MARCO. With T5-ColBERT, even if we use the top-$k'$ score for the imputation, the performance

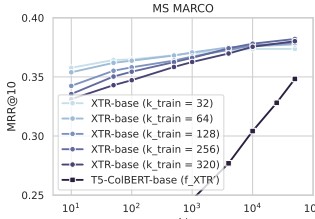 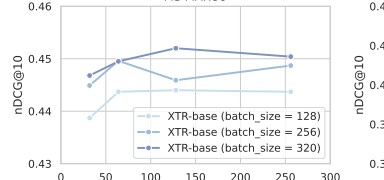 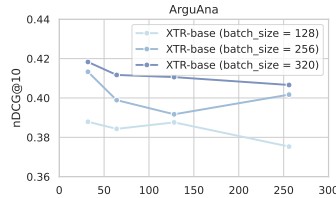

Figure 6: MRR@10 of XTR with different $k_{\text{train}}$ and $k'$. For T5-ColBERT, we also use $f_{\text{XTR}'}$ with the top-$k'$ score imputation method for the inference.

Figure 7: Effect of training XTR with different batch sizes and $k_{\text{train}}$. For each point of the graph, we train XTR$_{\text{base}}$ with the specified training batch size (128, 256, 320) and $k_{\text{train}}$ (32, 64, 128, 256) and evaluate on each dataset (MS MARCO and ArguAna). nDCG@10 of each model is reported.

is much worse than the original sum-of-max scoring. With XTR, the performance greatly improves as it has better token retrieval. Figure 5 shows how Recall@100 improves with larger $k'$'s as it provides more exact upper bound for the missing similarity imputation. Table D.2 shows that even if we use smaller $k'$, XTR still maintains high performances on BEIR.

### 5.3 Relationship between Hyperparameters

**$k_{\text{train}}$ vs. $k'$**   In Figure 6, we show MRR@10 of XTR trained with different $k_{\text{train}}$ and evaluated with different $k'$ on the MS MARCO development set. While all variants of XTR prefer larger $k'$, ones trained with smaller $k_{\text{train}}$ show higher performances than others under small $k'$ settings. XTR with larger $k_{\text{train}}$ exhibits better performances than ones with smaller $k_{\text{train}}$ as $k'$ becomes larger.

**Training batch size vs. $k_{\text{train}}$**   In Figure 7, we show the relationship between the training batch size and $k_{\text{train}}$ during training XTR. In this experiment, we use $k' = 40,000$. While it is evident that XTR mostly favors large training batch sizes, the optimal top-$k_{\text{train}}$ can be different for different datasets. While most datasets including MS MARCO favored a large enough $k_{\text{train}}$, ArguAna prefers smaller $k_{\text{train}}$. We hypothesize that this is due to the longer query length in ArguAna, which makes multi-vector models fall short compared to dual-encoders (see GTR vs. T5-ColBERT in Table 2).

### 5.4 Qualitative Analysis

Table 6 shows a prediction sample from MS MARCO. For T5-ColBERT, all of the top retrieved tokens are exact lexical matches. Surprisingly, none of the retrieved passages are about the query, demonstrating T5-ColBERT's failure to retrieve tokens from the correct context. In contrast, XTR retrieves fewer exact lexically matching tokens, but the contexts of the retrieved tokens are much more related to the query. This example explains the lower lexical token retrieval probability of XTR compared to T5-ColBERT in Figure 4 (bottom), but higher gold token retrieval performance in Figure 4 (top). For more qualitative examples, please see Appendix E.

## 6   Related Work

One of the main limitations of dense retrieval models is that encoding the query and document into a single vector constrains the representational power of the models. Polyencoder [Humeau et al., 2020], MEBERT [Luan et al., 2021], and MVR [Zhang et al., 2022a] propose to use multiple embeddings, instead of one, to represent the query or the document. A more recent approach is token-level multi-vector retrieval, which stores and retrieves with every token embedding. ColBERT [Khattab and Zaharia, 2020] is probably the most renowned model in this family. ALIGNER (i.e. T5-ColBERT) [Qian et al., 2022] extends ColBERT by scaling up the backbone langauge model and studying various strategies for aggregating the token-level alignment scores. These token-level retrieval models show strong effectiveness and out-of-domain generalization ability.

Efforts for reducing serving costs of multi-vector models have been mostly focused on the token-level retrieval stage. COIL [Gao et al., 2021] accelerates token-level retrieval by confining retrieval within exact match tokens, sharing the spirit of classic inverted indexing. CITADEL [Li et al., 2022] relaxes COIL with a lexical routing mechanism where a query token vector only retrieves from a subset of

**T5-ColBERT** token retrieval for *"what is the **usual** pay for stock associates at michael?"*

| Rank | Token | Context of Token | Relevance |
|------|-------|------------------|-----------|
| 1 | usual | routine passport services: the **usual** waiting time in logan to get your passport is four (4) to eight (8) weeks for routine applications. | No |
| 2 | usual | the **usual** pay days are the 1st and 16th of each month. for annual educational paraprofessionals there is no payroll lag. | No |
| 5 | usual | the **usual** part xiii tax rate is 25% (unless a tax treaty between canada and your home country reduces the rate). | No |
| 50 | usual | this is where one can challenge the judgment debtor's claim. one option creditors have is to try and make a deal with the debtor to take less than 25% (the **usual** amount of a wage levy). | No |
| 100 | usual | the **usual** maximum inventory is 1 talisman, 26 elemental runes, and 26 pure essence. the ingredients must be brought to an opposing altar ... from the runes being crafted. | No |

**XTR** token retrieval for *"what is the **usual** pay for stock associates at michael?"*

| Rank | Token | Context of Token | Relevance |
|------|-------|------------------|-----------|
| 1 | usual | store manager. 1 salary: the **usual** salary a store manager receives can be anywhere around $52,000 to $115,000 annually. | No |
| 2 | usual | 1 salary: the **usual** salary a store manager receives can be anywhere around $52,000 to $115,000 annually. 2 bonuses: publix provide bonuses that could reach up to $40,000. | No |
| 5 | average | **average** salaries for michaels stores stock associate: $9. michaels stores hourly pay trends based on salaries posted anonymously by michaels stores employees. | **Yes** |
| 50 | v | i think the a**v**g starting pay is closer to 30k for asst mgr trainees. it is an hourly position until you are fully trained (40 hours per week). | No |
| 100 | average | **average** macys salaries. the average salary for macys jobs is $32,000. average macys salaries can vary greatly due to company, location, industry, experience and benefits. | No |

Table 6: Token retrieval example from MS MARCO. Among the top 100 retrieved tokens, 100% of T5-ColBERT tokens are lexically identical as the query token `usual` while only 8% of XTR tokens are lexically identical. XTR retrieves the relevant passage by retrieving `average` for `usual`.

document token vectors routed to the same key. PLAID [Santhanam et al., 2022a] optimizes the speed of ColBERT by pruning weaker candidates in the earlier stages of retrieval and using better vector quantization. ColBERT-v2 [Santhanam et al., 2022b] further adopts residual representations with cluster centroids to improve the efficiency of ColBERT. On the other hand, how to accelerate the scoring stage remains under-explored. To the best of our knowledge, XTR is the first work to simplify the scoring stage and remove the gathering stage in multi-vector retrieval.

## 7 Conclusion

Multi-vector retrieval leverages query and document token representations for effective information retrieval. In this paper, we propose XTR that simplifies the existing three-stage inference of multi-vector models by improving the initial token retrieval stage. Specifically, XTR scores documents solely based on the retrieved tokens, which is also optimized during training with in-batch document tokens. As a result, XTR achieves state-of-the-art performances on zero-shot information retrieval benchmarks while greatly reducing the FLOPs of the scoring stage. We further show that our objective function indeed encourages better token retrieval, retrieving more tokens from gold documents, whose contexts are better aligned with the query.

## Limitations

In most of our experiments, XTR was trained on MS MARCO, a large-scale retrieval dataset in English. While our experiments were conducted in a fair setting where most baseline models also utilize MS MARCO, future use cases might need to remove its dependency on MS MARCO due to the license or language-specific issue. We believe that LLM-based retrieval dataset generation [Dai et al., 2022] would be able to mitigate the problem in the future.

## Acknowledgements

We would like to thank the anonymous reviewers for their helpful feedback. We also thank Nicholas Monath, Raphael Hoffmann, Kelvin Guu, Slav Petrov, and others at Google DeepMind for their helpful comments and discussion.

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

# A Derivatives w.r.t. Similarity Scores

**Sum-of-max** Here, we use a cross-entropy loss $\mathcal{L}_{\mathrm{CE}}$ with the sum-of-max operator $f_{\mathrm{ColBERT}}$ and analyze the derivatives with respect to the token similarity scores.

$$\mathcal{L}_{\mathrm{CE}} = -\log \frac{\exp f(Q, D^+)}{\sum_{b=1}^{B} \exp f(Q, D_b)} = -f_{\mathrm{ColBERT}}(Q, D^+) + \log \sum_{b=1}^{B} \exp f_{\mathrm{ColBERT}}(Q, D_b) \quad (5)$$

$$f_{\mathrm{ColBERT}}(Q, D) = \frac{1}{n} \sum_{i=1}^{n} \sum_{j=1}^{m} \mathbf{A}_{ij} \mathbf{P}_{ij} = \frac{1}{n} \sum_{i=1}^{n} \mathbf{P}_{i\hat{j}} \quad (6)$$

Here, we denote $\hat{j}$ as the index of the row-wise maximum value, dependent on each $i$ (i.e., $\mathbf{A}_{ij} = 1$). Given the cross-entropy loss with the sum-of-max operator, we compute the gradient with respect to one of the maximum token similarities $\mathbf{P}_{i\hat{j}}^+$ for a positive document $D^+ \in D_{1:B}$:

$$\begin{aligned}
\frac{\partial \mathcal{L}_{\mathrm{CE}}}{\partial \mathbf{P}_{i\hat{j}}^+} &= -\frac{f(Q, D^+)}{\partial \mathbf{P}_{i\hat{j}}^+} + \frac{1}{\sum_{b=1}^{B} \exp f(Q, D_b)} \frac{\partial}{\partial \mathbf{P}_{i\hat{j}}^+} \sum_{b=1}^{B} \exp f(Q, D_b) \\
&= -\frac{\partial}{\partial \mathbf{P}_{i\hat{j}}^+} \frac{1}{n} \sum_{i=1}^{n} \max_{1 \le j \le m} \mathbf{P}_{ij}^+ + \frac{1}{\sum_{b=1}^{B} \exp f(Q, D_b)} \sum_{b=1}^{B} \frac{\partial}{\partial \mathbf{P}_{i\hat{j}}^+} \exp f(Q, D_b) \\
&= -\frac{1}{n} + \frac{1}{\sum_{b=1}^{B} \exp f(Q, D_b)} \sum_{b=1}^{B} \exp f(Q, D_b) \frac{\partial f(Q, D_b)}{\partial \mathbf{P}_{i\hat{j}}^+} \\
&= -\frac{1}{n} + \frac{1}{n} \frac{\exp f(Q, D^+)}{\sum_{b=1}^{B} \exp f(Q, D_b)} = -\frac{1}{n}[1 - P(D^+|Q, D_{1:B})].
\end{aligned}$$

Similarly, the gradient w.r.t. a maximum token similarity $\mathbf{P}_{i\hat{j}}^-$ for a negative document $D^- \in D_{1:B}$ is computed as follows:

$$\begin{aligned}
\frac{\partial \mathcal{L}_{\mathrm{CE}}}{\partial \mathbf{P}_{i\hat{j}}^-} &= -\frac{f(Q, D^+)}{\partial \mathbf{P}_{i\hat{j}}^-} + \frac{1}{\sum_{b=1}^{B} \exp f(Q, D_b)} \frac{\partial}{\partial \mathbf{P}_{i\hat{j}}^-} \sum_{b=1}^{B} \exp f(Q, D_b) \\
&= \frac{1}{n} \frac{\exp f(Q, D^-)}{\sum_{b=1}^{B} \exp f(Q, D_b)} = \frac{1}{n} P(D^-|Q, D_{1:B}).
\end{aligned}$$

Hence, the positive token-level score $\mathbf{P}_{i\hat{j}}^+$ will gradually increase until $P(D^+|Q, D_{1:B}) \to 1$ and the negative token-level score $\mathbf{P}_{i\hat{j}}^-$ will decrease until $P(D^-|Q, D_{1:B}) \to 0$. This shows that the token-level scores are trained based on the document-level scores, which might stagnate the token-level scores. For instance, even if $\mathbf{P}_{i\hat{j}}^-$ is very high—later causing $\mathbf{d}_{\hat{j}}^-$ to be retrieved instead of ones from positive documents—it will not be penalized as long as $P(D^-|Q, D_{1:B})$ is low enough.

**In-batch token retrieval** Compared to the sum-of-max operator, our in-batch sum-of-max $f_{\mathrm{XTR}}$ considers the max values only when they are retrieved over other negative tokens in the mini-batch.

$$f_{\mathrm{XTR}}(Q, D_{1:B}) = \frac{1}{Z} \sum_{i=1}^{n} \sum_{j=1}^{m} \mathbf{A}_{ij} \hat{\mathbf{A}}_{ij} \mathbf{P}_{ij} = \frac{1}{Z} \sum_{i=1}^{n} \mathbf{P}_{i\bar{j}}$$

Here, we denote $\bar{j}$ as the index of the row-wise maximum value that is also within the mini-batch top-$k_{\mathrm{train}}$ given $q_i$ (i.e., satisfies both $\mathbf{A}_{ij} = 1$ and $\hat{\mathbf{A}}_{ij} = 1$). If there is no such $\bar{j}$, we simply use $\mathbf{P}_{i\bar{j}} = 0$. We also use a normalizer $Z$, which is the number of non-zero $\mathbf{P}_{i\bar{j}}$. In this analysis, we assume $Z > 0$ since if every $\mathbf{P}_{i\bar{j}}$ is zero, the gradient is undefined.

The gradient w.r.t. the maximum token similarity $\mathbf{P}^+_{i\hat{j}}$ (non-zero) for a positive document $D^+ \in D_{1:B}$ is computed as follows:

$$\frac{\partial \mathcal{L}_{\text{CE}}}{\partial \mathbf{P}^+_{i\hat{j}}} = -\frac{f(Q, D^+)}{\partial \mathbf{P}^+_{i\hat{j}}} + \frac{1}{\sum_{b=1}^B \exp f(Q, D_b)} \frac{\partial}{\partial \mathbf{P}^+_{i\hat{j}}} \sum_{b=1}^B \exp f(Q, D_b)$$

$$= -\frac{1}{Z^+}\Big[1 - \frac{\exp f(Q, D^+)}{\sum_{b=1}^B \exp f(Q, D_b)}\Big]$$

$$= -\frac{1}{Z^+}\big[1 - P(D^+|Q, D_{1:B})\big].$$

This is a very similar result compared to the sum-of-max operator except that 1) the gradient is defined only when $\mathbf{P}^+_{i\hat{j}}$ is non-zero (i.e. retrieved) and 2) it is dependent on $Z^+$, which means that the gradient will be large whenever there is a small number of retrieved tokens from the positive document. If only a handful of tokens are retrieved for $D^+$, our objective function increases $\mathbf{P}^+_{i\hat{j}}$.

For negative similarity score $\mathbf{P}^-_{i\hat{j}}$, we have the following:

$$\frac{\partial \mathcal{L}_{\text{CE}}}{\partial \mathbf{P}^-_{i\hat{j}}} = -\frac{f(Q, D^+)}{\partial \mathbf{P}^-_{i\hat{j}}} + \frac{1}{\sum_{b=1}^B \exp f(Q, D_b)} \frac{\partial}{\partial \mathbf{P}^-_{i\hat{j}}} \sum_{b=1}^B \exp f(Q, D_b)$$

$$= -\frac{1}{Z^-}\Big[-\frac{\exp f(Q, D^-)}{\sum_{b=1}^B \exp f(Q, D_b)}\Big]$$

$$= \frac{1}{Z^-} P(D^-|Q, D_{1:B}).$$

Again, it is similar to the sum-of-max result, but it depends on $Z^-$. In this case, even when $P(D^-|Q, D_{1:B})$ is low, if there is a small number of retrieved tokens from $D^-$ (i.e., small $Z^-$), $\mathbf{P}^-_{i\hat{j}}$ will be decreased significantly. Note that when $Z^-$ is large, $Z^+$ naturally becomes smaller as they compete for in-batch token retrieval, which causes positive tokens to have higher scores.

## B  Inference Complexity

We compare the complexity of ColBERT and XTR during the scoring stage in terms of FLOPs. We do not measure the complexity for the online query encoding and maximum inner product search (MIPS), which have been extensively studied for both dual encoders and multi-vector retrieval [Santhanam et al., 2022a,b, Guo et al., 2020].

For the scoring stage, both ColBERT and XTR have $\mathcal{O}(nk')$ candidate documents. Here, we assume the worst case $nk'$ where each document token comes from a unique document. For each candidate document, ColBERT loads a set of document vectors of $\bar{m}d$ floating points ($\bar{m}$ = average document length) and computes eq. (1) with the query vectors of $nd$ floating points. Computing eq. (1) per candidate document requires $2n\bar{m}d$ FLOPs for token-level inner products, $n\bar{m}$ for finding the row-wise max, and $n$ for the final average. In total, ColBERT requires $n^2 k'(2\bar{m}d + \bar{m} + 1)$ FLOPs for the scoring stage. Note that this does not include the latency of loading the $\mathcal{O}(nk'\bar{m}d)$ floating points onto the memory, which amounts up to 450MB per query when $n = 16, k' = 1000, \bar{m} = 55, d = 128$.

On the other hand, XTR first imputes the missing similarity, which is simply done by caching the $k'$-th token retrieval score for each query token. Then, each of $nk'$ candidate documents requires $n\bar{r}$ FLOPs for finding row-wise max and $n$ for the average where $\bar{r}$ is the average number of retrieved tokens per each candidate document. In total, we have $n^2 k'(\bar{r} + 1)$ FLOPs. Table 1 shows the estimated FLOPs of the two models. XTR reduces the FLOPs at the scoring stage by 4000× making multi-vector retrieval more efficient and practical.

# C    Implementation Details

XTR uses $k_{\text{train}}$ for retrieving in-batch document tokens. Since we retrieve over mini-batches, the size of mini-batch affects the performance for different $k_{\text{train}}$, which is shown in §5.3. In our experiments, we tried $k_{\text{train}} = \{32, 64, 128, 256, 320\}$ for each batch size and choose the best model based on their performance on the MS MARCO development set. For inference, XTR uses $k'$ for the token retrieval. We use $k' = 40,000$, which is possible due to the efficient scoring stage of XTR.[6] We analyze the effect of using different $k'$'s as well as its relationship to $k_{\text{train}}$ in §5.3. We initialize XTR from the base and xxl versions of the T5 encoder [Raffel et al., 2020] and provide $XTR_{\text{base}}$ and $XTR_{\text{xxl}}$. For multilingual XTR, we initialize XTR from mT5 [Xue et al., 2021]. We fine-tune XTR for 50,000 iterations with the learning rate to 1e-3. Up to 256 chips of TPU v3 accelerator were used depending on the size of the model. We use ScaNN [Guo et al., 2020] for the MIPS during the token retrieval stage. For BEIR, we use 13 datasets (AR: ArguAna. TO: Touché-2020. FE: Fever. CF: Climate-Fever. SF: Scifact. CV: TREC-COVID. NF: NFCorpus. NQ: Natural Questions. HQ: HotpotQA. FQ: FiQA-2018. SD: SCIDOCS. DB: DBPedia. QU: Quora).

**Baselines**    There are two main paradigms on training retriever models for the out-of-domain evaluation. The first group trains a single retriever for each dataset (or domain) by generating queries for each out-of-domain corpus. Typically, this approach generates $N$ datasets to train $N$ independent models for $N$ different domains. For this *one-retriever-per-domain* approaches, we include GenQ [Thakur et al., 2021], GPL [Wang et al., 2022], and Promptagator [Dai et al., 2022]. The second group builds a single retriever—typically trained on a large-scale IR dataset such as MS MARCO—and directly applies it on the out-of-domain corpora and queries. For this *one-retriever-for-all* approaches, we present results of state-of-the-art retrievers including $Splade_{v2}$ [Formal et al., 2021], $ColBERT_{v2}$ [Santhanam et al., 2022b], and $GTR_{xxl}$ [Ni et al., 2021]. We also show the results of $T5\text{-}ColBERT_{xxl}$ [Qian et al., 2022], which is a T5-initialized ColBERT model and shares the same backbone LM and training dataset with XTR. Note that T5-ColBERT uses the heavy scoring stage based on the original sum-of-max. All of our one-retriever-for-all baselines, as well as XTR, are trained on English MS MARCO, unless otherwise stated.

---

[6]In fact, XTR with $k' = 40,000$ has still two-to-three orders of magnitude cheaper scoring stage than ColBERT with $k' = 1,000$ and T5-ColBERT with $k' = 4,000$.

# D   Additional Results

In Table D.1, we show Recall@100 on BEIR.

| | MS | AR | TO | FE | CF | SF | CV | NF | NQ | HQ | FQ | SD | DB | QU | Avg. |
|---|---|---|---|---|---|---|---|---|---|---|---|---|---|---|---|
| | | | | | | *One Retriever per Domain* | | | | | | | | | |
| GenQ | 88.4 | 97.8 | 45.1 | 92.8 | 45.0 | 89.3 | 45.6 | 28.0 | 86.2 | 67.3 | 61.8 | 33.2 | 43.1 | 98.9 | 64.2 |
| $PTR_{retriever}$ | - | 98.9 | 47.5 | 94.1 | 53.1 | 91.8 | 55.9 | 30.6 | 89.8 | 74.6 | 76.5 | 41.6 | 46.3 | 99.6 | 69.2 |
| | | | | | | *One Retriever for All* | | | | | | | | | |
| BM25 | 65.8 | 94.2 | 53.8 | 93.1 | 43.6 | 90.8 | 49.8 | 25.0 | 76.0 | 74.0 | 53.9 | 35.6 | 39.8 | 97.3 | 63.6 |
| ColBERT | 86.5 | 91.4 | 43.9 | 93.4 | 44.4 | 87.8 | 46.4 | 25.4 | 91.2 | 74.8 | 60.3 | 34.4 | 46.1 | 98.9 | 64.5 |
| $GTR_{base}$ | 89.8 | 97.4 | 44.3 | 92.3 | 52.2 | 87.2 | 41.1 | 27.5 | 89.3 | 67.6 | 67.0 | 34.0 | 41.8 | 99.6 | 64.7 |
| $T5\text{-}ColBERT_{base}$ | 91.8 | 76.0 | 49.9 | 90.4 | 46.2 | 91.3 | 55.4 | 27.6 | 90.5 | 78.3 | 63.0 | 34.2 | 50.5 | 97.9 | 65.5 |
| **$XTR_{base}$** | 91.0 | 92.1 | 50.8 | 92.5 | 51.6 | 90.5 | 57.3 | 28.0 | 91.6 | 80.7 | 63.5 | 34.8 | 52.0 | 98.9 | 68.0 |
| $GTR_{xxl}$ | 91.6 | 98.3 | 46.6 | 94.7 | 55.6 | 90.0 | 40.7 | 30.0 | 94.6 | 75.2 | 78.0 | 36.6 | 49.4 | 99.7 | 68.4 |
| $T5\text{-}ColBERT_{xxl}$ | 93.3 | 81.4 | 50.1 | 91.7 | 49.8 | 94.6 | 60.3 | 29.0 | 95.5 | 81.6 | 72.5 | 38.5 | 54.6 | 99.1 | 69.1 |
| **$XTR_{xxl}$** | 93.0 | 95.6 | 52.7 | 93.7 | 56.2 | 95.0 | 62.1 | 30.7 | 95.8 | 82.2 | 73.0 | 39.4 | 54.5 | 99.3 | **71.6** |

Table D.1: Recall@100 on MS-MARCO and BEIR. The last column shows the average over 13 BEIR benchmarks. Compared to GTR, T5-ColBERT only marginally improves the recall. On the other hand, XTR greatly improves the recall showing the importance of having a better token retrieval.

In Table D.2, we show nDCG@10 and Recall@100 on BEIR with different $k'$.

| $k'$ | MS | AR | TO | FE | CF | SF | CV | NF | NQ | HQ | FQ | SD | DB | QU | Avg. |
|---|---|---|---|---|---|---|---|---|---|---|---|---|---|---|---|
| | | | | | | **nDCG@10** | | | | | | | | | |
| 40,000 | **45.0** | 40.7 | **31.3** | **73.7** | **20.7** | 71.0 | **73.6** | 34.0 | **53.0** | **64.7** | **34.7** | 14.5 | **40.9** | 86.1 | **49.1** |
| 1,000 | 43.2 | **44.6** | 29.0 | 72.1 | 20.4 | **71.7** | 67.5 | **34.2** | 49.8 | 61.3 | 33.0 | **15.9** | 37.0 | **86.3** | 47.9 |
| | | | | | | **Recall@100** | | | | | | | | | |
| 40,000 | **91.0** | 92.1 | **50.8** | 92.5 | 51.6 | 90.5 | **57.3** | 28.0 | **91.6** | 80.7 | **63.5** | 34.8 | **52.0** | 98.9 | **68.0** |
| 1,000 | 88.8 | **96.4** | 48.0 | **92.5** | **53.3** | **93.1** | 48.1 | **28.6** | 88.8 | 78.3 | 62.5 | **37.0** | 47.0 | **99.1** | 67.1 |

Table D.2: nDCG@10 and Recall@100 of $XTR_{base}$ on MS-MARCO and BEIR with different $k'$. The last column shows the average over 13 BEIR benchmarks.

# E  Qualitative Analysis

In Table 6-E.5, we show token retrieval results from T5-ColBERT and XTR.

| Rank | Token | Context of Token | Relevance |
|------|-------|------------------|-----------|
| **T5-ColBERT** token retrieval for "*lauren london age*?" | | | |
| 1 | la | **la**ura bush laura lane welch bush (born november 4, 1946) is the wife of the 43rd president of the united states, george w. bush. | No |
| 2 | la | is laura branigan dead? **la**ura branigan died on august 26, 2004 at the age of 47. | No |
| 5 | la | laika death in space. **la**ika died within hours from overheating. her body temperature got way too hot for her to survive. the heat in her spacecraft had risen to 40 degrees celsius (104 degrees fahrenheit). | No |
| 50 | la | singer **la**ura branigan dies at 47 singer laura branigan dies at 47. laura branigan, a grammy-nominated pop singer best known for her 1982 platinum hit gloria, has died. | No |
| 100 | la | **la**uren bacall lauren bacall ( born betty joan perske; september 16, 1924 august) | No |

| Rank | Token | Context of Token | Relevance |
|------|-------|------------------|-----------|
| **XTR** token retrieval for "*lauren london age*?" | | | |
| 1 | la | lauren london birthday, age, family & biography 33 years, 1 month, 23 days old age **la**uren london will turn 34 on 05 december, 2018. | Yes |
| 2 | la | **la**uren london current age 33 years old. lauren london height 5 feet 7 inches (1.5 m/ 157 cm) and her weight 119 lbs (54 kg). | Yes |
| 5 | la | until now, **la**uren taylor's age is 28 year old and have gemini constellation. count down 363 days will come next birthday of lauren taylor! | No |
| 50 | la | if dwayne johnson, 43, and his longtime girlfriend, **la**uren hashian, 31, have a baby, would they have a pebble? the furious 7 star and his bae are reportedly expecting their first child together. | No |
| 100 | la | laura bush biography after his defeat, bush returned to is oil business and **la**ura became a housewife, but soon returned to politics to help her father-in-law, george h.w. bush's presidential campaign in 1980. | No |

Table E.1: Token retrieval example from MS MARCO for the token *"la"* in the query *"lauren london age"*. Among the top 100 retrieved tokens, $100\%$ of T5-ColBERT tokens are lexically identical as the query token `la` and $100\%$ of XTR tokens are also lexically identical. However, top retrieved results from XTR contain the correct entity (`Lauren London`) while those from T5-ColBERT are about wrong entities (`Laura Bush`, `Laura Branigan`, etc.).

| **T5-ColBERT** token retrieval for "*temple university student population*?" | | | |
|---|---|---|---|
| Rank | Token | Context of Token | Relevance |
| 1 | temple | about **temple** university tuition, cost, financial aid, scholarships, and admission rates | No |
| 2 | temple | overview the application fee at **temple** university is $55. it is selective, with an acceptance rate of 61.7 percent and an early acceptance rate of 78 percent. | No |
| 5 | temple | the application fee at **temple** university is $55. it is selective, with an acceptance rate of 61.7 percent and an early acceptance rate of 78 percent. | No |
| 50 | temple | **temple** university staff accountants earn $52,000 annually, or $25 per hour, which is 14% higher than the national average for all staff accountants at $45,000 annually and 16% lower than the national salary average for all working americans | No |
| 100 | temple | browse expedia's selection and check out the best hotels close to **temple** university for the world-class spas and restaurants, or snatch up one of the cheap hotel deals near temple university | No |

| **XTR** token retrieval for "*temple university student population*?" | | | |
|---|---|---|---|
| Rank | Token | Context of Token | Relevance |
| 1 | temple | by gender, the school has 18,009 male and 19,476 female students. by race/ethnicity, 20,664 white, 4,466 black, and 3,819 asian students are attending at **temple** university. | Yes |
| 2 | temple | below tables and charts represent the enrollment statistics including school degree, gender, race/ethnicity, and tranfer-in students at the school. at **temple** university, 37,485 students are enrolled .... | Yes |
| 5 | temple | temple university the big picture: how many students were on campus in fall 2015? of the 28,886 new freshman applicants, 56% were admitted and 31% of the admitted students enrolled at **temple** university in fall 2015. | Yes |
| 50 | temple | **temple** university was founded in 1884 by russell conwell, a yale-educated boston lawyer, orator, and ordained baptist minister | No |
| 100 | temple | kaiser said **temple**'s endowment fund is low because the university is late to the idea of fundraising. | No |

Table E.2: Token retrieval example from MS MARCO for the token *"temple"* in the query *"temple university student population?"*. Among the top 100 retrieved tokens, $100\%$ of T5-ColBERT tokens are lexically identical as the query token temple and $100\%$ of XTR tokens are also lexically identical. However, top retrieved results from XTR are of the correct context (student population) while those from T5-ColBERT are off-topic (e.g., tuition, salary, etc.).

**T5-ColBERT** token retrieval for *"**aire** is expressed in some skin tumors"*

| Rank | Token | Context of Token | Relevance |
|---|---|---|---|
| 1 | aire | acids: structures, properties, and functions (university science books, sausalito, ca, 2000). humans expressing a defective form of the transcription factor **aire** (autoimmune regulator) develop multiorgan autoimmune disease. | No |
| 2 | aire | the primary biochemical defect in apeced is unknown. we have isolated a novel gene, **aire**, encoding for a putative nuclear protein featuring two phd-type zinc-finger motifs, suggesting its involvement in transcriptional regulation. | No |
| 5 | aire | control of central and peripheral tolerance by **aire**. the negative selection of self-reactive thymocytes depends on the expression of tissue-specific antigens by medullary thymic epithelial cells. | No |
| 50 | aire | we found that a human patient and mice with defects in aire develop similar lung pathology, demonstrating that the **aire**-deficient model of autoimmunity is a suitable translational system in which to unravel fundamental mechanisms of ild pathogenesis. | No |
| 100 | air | cool **air** initiates just downstream of the major sense transcript poly(a) site and terminates either early or extends into the flc promoter region. | No |

**XTR** token retrieval for *"**aire** is expressed in some skin tumors"*

| Rank | Token | Context of Token | Relevance |
|---|---|---|---|
| 1 | aire | keratin-dependent regulation of **aire** and gene expression in skin tumor keratinocytes expression of the intermediate filament protein keratin 17 (k17) is robustly upregulated in inflammatory skin diseases and in many tumors.... | Yes |
| 2 | aire | the thymic transcription factor autoimmune regulator (aire) prevents autoimmunity in part by promoting expression of tissue-specific self-antigens, which include many cancer antigens. for example, **aire**-deficient patients are predisposed to vitiligo, an autoimmune disease of melanocytes that is often triggered by efficacious immunotherapies against melanoma. | Yes |
| 5 | aire | aire regulates negative selection of organ-specific t cells autoimmune polyendocrinopathy syndrome type 1 is a recessive mendelian disorder resulting from mutations in a novel gene, **aire**, and is characterized by a spectrum of organ-specific autoimmune diseases. | No |
| 50 | aire | here we demonstrate a novel role for a cd4+3- inducer cell population, previously linked to development of organized secondary lymphoid structures and maintenance of t cell memory in the functional regulation of **aire**-mediated promiscuous gene expression in the thymus. | No |
| 100 | air | this localization is dependent on the presence of sperm in the spermatheca. after fertilization, **air**-2 remains associated with chromosomes during each meiotic division. | No |

Table E.3: Token retrieval example from MS MARCO for the token *"aire"* in the query *"aire is expressed in some skin tumors"*. Among the top 100 retrieved tokens, 77% of T5-ColBERT tokens are lexically identical as the query token `aire` and 77% of XTR tokens are also lexically identical. Top retrieved results from XTR are relevant to the query (about `cancer`, `tumor`, `skin`, and `melanocyte`), while those from T5-ColBERT are off-topic.

**T5-ColBERT** for *"women with a higher birth weight are more likely to develop breast cancer later in life"*

| Rank | Token | Context of Token | Relevance |
|------|-------|------------------|-----------|
| 1 | later | context exposure to cardiovascular risk factors during childhood and adolescence may be associated with the development of atherosclerosis later in life. | No |
| 2 | later | n despite the high incidence of febrile seizures, their contribution to the development of epilepsy later in life has remained controversial. | No |
| 5 | later | prospectively collected data from two intervention studies in adults with severe malaria were analysed focusing on laboratory features on presentation and their association with a later requirement for rrt. | No |
| 50 | later | they did have a limited amount of proteolytic activity and were able to kill s. aureus. with time, the nuclear envelope ruptured, and dna filled the cytoplasm presumably for later lytic net production | No |
| 100 | late | finally, we address the need for a careful consideration of potential benefits of bisphosphonate therapy and the risk for osteonecrosis of the jaw, a recently recognized late-toxicity of their use. | No |

**XTR** for *"women with a higher birth weight are more likely to develop breast cancer later in life."*

| Rank | Token | Context of Token | Relevance |
|------|-------|------------------|-----------|
| 1 | later | life course breast cancer risk factors and adult breast density (united kingdom) objective to determine whether risk factors in childhood and early adulthood affect later mammographic breast density. | Yes |
| 2 | later | exposure to cardiovascular risk factors during childhood and adolescence may be associated with the development of atherosclerosis later in life. | No |
| 5 | subsequent | emerging evidence suggests an association between female prenatal experience and her subsequent risk of developing breast cancer. | Yes |
| 50 | later | our nested case–control study of eh progression included 138 cases, who were diagnosed with eh and then with carcinoma (1970–2003) at least 1 year (median, 6.5 years) later, and 241 controls.... | No |
| 100 | during | obesity and being overweight during adulthood have been consistently linked to increased risk for development of dementia later in life, especially alzheimer's disease. | No |

Table E.4: Token retrieval example from Scifact for the token *"later"* in the query *"women with a higher birth weight are more likely to develop breast cancer later in life"*. Among the top 100 retrieved tokens, 72% of T5-ColBERT tokens are lexically identical as the query token `later` while only 33% of XTR tokens are lexically identical. Top retrieved results from XTR can retrieves synonyms (`sebsequent`) from relevant context, while those from T5-ColBERT are off-topic.

**T5-ColBERT** for *"venules have a **thinner** or absent smooth layer compared to arterioles."*

| Rank | Token | Context of Token | Relevance |
|------|-------|------------------|-----------|
| 1 | **thinner** | platelet cd40l is associated with smaller plaques and **thinner** caps, while p-selectin is associated with smaller core size. conclusions: blood cell activation is significantly associated with atherosclerotic changes of the carotid wall. | No |
| 2 | **thin** | the periosteum is a **thin**, cellular and fibrous tissue that tightly adheres to the outer surface of all but the articulated surface of bone and appears to play a pivotal role in driving fracture pain. | No |
| 5 | **thin** | immunohistological scoring showed significantly (p<0.0001) higher median 5hmc levels in bcn and dcn than in **thin** ssm, thick ssm, and cmd. | No |
| 50 | **weak** | subarachnoid haemorrhage (1·43 [1·25-1·63]), and stable angina (1·41 [1·36-1·46]), and **weak**est for abdominal aortic aneurysm (1·08 [1·00-1·17]). | No |
| 100 | **slight** | the ucp-2 gene expression was widely detected in the whole body with substantial levels in the wat and with **slight** levels in the skeletal muscle and bat. | No |

**XTR** for *"venules have a **thinner** or absent smooth layer compared to arterioles."*

| Rank | Token | Context of Token | Relevance |
|------|-------|------------------|-----------|
| 1 | **thinner** | platelet cd40l is associated with smaller plaques and **thinner** caps, while p-selectin is associated with smaller core size. conclusions: blood cell activation is significantly associated with atherosclerotic changes of the carotid wall. | No |
| 2 | **thin** | the periosteum is a **thin**, cellular and fibrous tissue that tightly adheres to the outer surface of all but the articulated surface of bone and appears to play a pivotal role in driving fracture pain. | No |
| 5 | **thick** | in dense fibrotic zones, **thick**ening of the arterial and venous wall with severe luminal narrowing was present in each patient. | No |
| 50 | **small** | we assessed vasomotor function of the adipose microvasculature using videomicroscopy of **small** arterioles isolated from different fat compartments. | No |
| 100 | **particle** | context circulating concentration of lipoprotein(a) (lp[a]), a large glycoprotein attached to a low-density lipoprotein-like **particle**, may be associated with risk of coronary heart disease (chd) and stroke. | No |

Table E.5: Token retrieval example from Scifact for the token *"thinner"* in the query *"vanules have a thinner or absent smooth later compared to arterioles"*. Among the top 100 retrieved tokens, only $1\%$ of T5-ColBERT tokens are lexically identical as the query token `thinner` and only $1\%$ of XTR tokens are also lexically identical.

