# OpenReview forum: "Rethinking the Role of Token Retrieval in Multi-Vector Retrieval"
_NeurIPS.cc/2023/Conference — NeurIPS 2023 poster_

### Official Review · Reviewer_XbZR · 2023-07-05

**Soundness:** 3 good
**Presentation:** 4 excellent
**Contribution:** 4 excellent
**Rating:** 7
**Confidence:** 4

**Summary:**

The authors improve multi-vector retrieval to move beyond the standard retrieve-gather-score stages of ColBERT. In particular, they modify the objective function during training as well as the scoring mechanism so it doesn't require gathering all token vectors of each candidate document before the final scores are computed.

The authors begin by showing that, in ColBERT, the standard cross-entropy loss applied over the aggregated scores fails to reduce token-level scores if the average score is low. This can reduce the precision of the initial "retrieve" stage, putting a higher burden on the gather/score stages.

To tackle that, the authors simulate the token retrieval stage during training by masking/skipping document token scores for tokens that aren't among the nearest top_{k_{train}} to a given query token within the training batch. At search time, the authors use exclusively the retrieved (i.e., close) tokens from candidate documents. Scores of "missing" tokens are imputed/estimated with a lower bound from the kNN retrieval stage.

**Strengths:**

The paper is very strong overall. It tackles a well-defined problem with a well-motivated and novel solution. I can see this being widely applicable to multi-vector / ColBERT-like retrievers. The analysis and rich results are very strong as well, considering the paper doesn't use distillation from cross encoders. (The paper does use hard negatives, though --- from RocketQA).

**Weaknesses:**

Despite the very strong contributions, a number of claims of the paper are needlessly inflated (or unsupported).

First, the abstract (and paper) say that XTR "enables a newly designed scoring stage that is two-to-three orders of magnitude cheaper than that of ColBERT". The basis of this claim is a theoretical FLOPs analysis of one of the three scoring stages of ColBERT against XTR.

To the best of my knowledge, this appears, however, unsupported or otherwise inflated on a few dimensions:

1) This comparison doesn't count the actual FLOPs used. It also doesn't measure the latency of the scoring stage or the full pipeline. Even though these measurements may be less idealized than theoretical analysis, not reporting them makes it much harder to assess the proposed methods.

2) While XTR appears to make the scoring stage essentially free (at least based on the FLOPs analysis), it is unclear how much this affects the total computational cost of retrieval. Is the latency now low? Is it lower than existing multi-vector retrievers? The paper offers no empirical insight on this, but it's unlikely (or, rather, impossible?) that XTR is 2-3 orders of magnitude faster than them overall. For instance, ColBERT retriever in PLAID (2022) seems to report latency of 58 ms (or lower) per query on MS MARCO. Two orders of magnitude faster than that would be 0.6 milliseconds per query, which is much faster than even basic BM25 retrieval permits.

3) The comparison is conducted against the original ColBERT (2020), but this would appear to ignore years of optimizations for multi-vector scoring as in ColBERTv2 (2021), PLAID (2022), and other papers like ColBERTer (2022), etc. For instance, the PLAID ColBERTv2 retriever appears to show that candidate scoring (and the corresponding index lookup) is an extremely cheap step. This may be because PLAID moves a lot of the work to earlier stages, making them more expensive instead(?). However, there is very minimal engagement in the paper with the concerns of these developments in terms of faster candidate generation and/or much smaller index. While this isn't the focus of the paper, it's important to (at least) report the index size, how much of it needs to reside in memory, and how fast the overall retrieval process can be for XTR.

Second, the abstract reports that "on the popular BEIR benchmark, XTR advances the state-of-the-art by 2.8 nDCG@10 without any distillation." It's true this is achieved without any distillation, but such a result is only achieved with the XXL (11B or 5.5B?) T5 encoder of XTR, which is orders of magnitude larger than related methods. This isn't an issue with the evaluation itself, but it shouldn't be glossed over.

**Questions:**

What is the intuition for the normalizer Z during training? How essential is this for the results?

How robust are the analyses with the T5-ColBERT, compared to the original ColBERT and ColBERTv2 models? ColBERT typically has a query expansion stage that appears to be skipped in XTR. How does this affect the analysis and results?

**Limitations:**

See Weaknesses & Questions.

---

> ### Author Rebuttal · Authors · 2023-08-08
>
> Thank you for your detailed review.
>
> **A1. This comparison doesn't count the actual FLOPs used. It also doesn't measure the latency of the scoring stage or the full pipeline.**
>
> As the reviewer summarized, our goal of the paper is to simplify the three-stage inference of ColBERT while making the scoring stage very efficient. Also, our claim on the FLOPs improvement is over the scoring stage, not over the entire pipeline. The apples-to-apples comparison of all baselines is difficult due to the differences in libraries (e.g. FAISS vs ScaNN, pytorch vs jax) and implementations depending on hardware constraints (e.g., multiprocessing, RAM usage). Nevertheless, for the sake of understanding the actual latency, we summarize the latency of our current implementation of XTR in the general response G1. We also show how much memory is needed to compute the full sum-of-max for T5-ColBERT. Please note that optimizing the latency of the entire pipeline is not our immediate goal as it requires lots of engineering efforts, which will depend on the given resources and algorithms. We will make this clearer in the paper. For comparisons against PLAID and ColBERTer, please see our general response G2.
>
> **A2. It's true this is achieved without any distillation, but such a result is only achieved with the XXL (11B or 5.5B?) T5 encoder of XTR, which is orders of magnitude larger than related methods.**
>
> In the abstract, we will properly tone down the state-of-the-art message and focus more on the contributions of XTR. We use the encoder part of T5-XXL so it is about 5B.
>
> **A3. What is the intuition for the normalizer Z during training? How essential is this for the results?**
>
> The normalizer Z provides stable losses during training. Since each document has a different number of mini-batch tokens retrieved, removing the normalization would give very high scores on positive documents, making the training process unstable. Not using Z (or setting Z to a constant) does not perform well compared to our proposed version of Z.
>
> **A4. How robust are the analyses with the T5-ColBERT, compared to the original ColBERT and ColBERTv2 models?**
>
> While most of our analyses were based on T5-ColBERT, we do acknowledge that there is a difference between T5-ColBERT and ColBERT (or v2) including query expansion and cross-encoder distillation. Based on our theoretical analyses in Appendix A, which uses the sum-of-max operator, we believe that our findings would hold for existing ColBERT variants even if they use the query expansion. More empirical comparisons against ColBERT could be needed to better support our hypothesis in the future.

---

> > ### Comment · Reviewer_XbZR · 2023-08-17
> >
> > Thank you for the response. I think XTR is a powerful contribution conceptually and from a modeling standpoint. Because of that, I am willing to keep my score if the authors promise to tone down two claims in the paper pertaining to:
> >
> > 1. [from A2] "we will properly tone down the state-of-the-art message and focus more on the contributions"
> >
> > 2. [from A3] "Not using Z (or setting Z to a constant) does not perform well compared to our proposed version of Z"
> >
> > 3. [from A1] "our goal of the paper is to simplify the three-stage inference of ColBERT while making the scoring stage very efficient. Also, our claim on the FLOPs improvement is over the scoring stage, not over the entire pipeline"
> >
> > To expand on this one, A1 is a perfectly reasonable goal, and I agree that this work has achieved that. However, the current abstract, intro, and writing may have left me with the impression that XTR was in fact verified to be faster than (or at least competitive with!) existing ColBERT implementations. If how XTR would interact with systems work on optimizing these models is not even considered, this should be very clear to the reader.
> >
> > In particular, the latency numbers the authors report in G1 are not competitive with any recent realistic implementation of ColBERT-like methods to my knowledge, which generally search in tens or hundreds of milliseconds at most. While I understand that "differences in libraries and infrastructure" are significant, it is critical to be clear in the intro that (1) the author's only implementation of XTR remains considerably less efficient in latency and in storage than existing work, (2) in the author's tests of latency the scoring phase is 7-8x faster under such and such simplified conditions (which certainly softens the 4000x FLOPs reduction claim), and (3) future work is needed to confirm that the XTR efficiency gains can be realized in practice in a way that complements existing work.

---

> > > ### Author Response · Authors · 2023-08-19
> > >
> > > Thank you for the insightful suggestions! We will tone down the claims in our paper. We will make it clear that the FLOPs improvement we reported is over a vanilla implementation of the scoring phase, which serves as a proof-of-concept. It remains to be studied how XTR can improve efficiency over highly optimized ColBERT implementations from prior work.

---

### Official Review · Reviewer_HcB5 · 2023-07-06

**Soundness:** 2 fair
**Presentation:** 1 poor
**Contribution:** 2 fair
**Rating:** 6
**Confidence:** 4

**Summary:**

Authors propose a better document retrieval method. They build on top of ColBert and instead of reranking using all tokens of documents retrieved by stage 1(a query token document token retrieval), they just use retrieved document tokens from stage 1 and perform retrieval using them.

**Strengths:**

-	Results seem to be strong enough.
-	Do a decent job in ablations, qualitative analysis, complexity analysis.

**Weaknesses:**

-	The paper overcomplicates a simple concept (Eq 4 can be further simplified from what I understand). Max over j would collapse then.
-	Given the proposed methodology is a training mechanism, it would be interesting to see how this method performs with training on NQ, TriviaQA, etc. Currently, the authors perform trainning only on MS MACRO and perform testing on NQ, triviaQA, etc. DPR does this.
-	Very bad writing.
-	Very handwavy at some places
-	See Questions for detailed list of weakness.

**Questions:**

-	For the choice of m_i, Eq. 4 would have collapsed to the form of Eq.1 with extra Ai. Why show it in this form and complicate? Isn't Max over j the same as Aij(qi.Tdj)??
-	Line 120. How is $f_{colbert}(Q, D^-)=0.2$ when $q_i^Td^-_j>0.8$. Failure case very handwavy. Please explain with proper numbers and reasoning. Is Fig2 different tokens for any +ve/-ve documents or just the retrieved tokens?
-	Lines 134-137. Very handwavy again. Authors should use proper math format to denote in-batch tokens. It is very difficult to understand when $A_{ij}=1$.
-	Lines 134-137. Define top-k_train
-	For the document tokens and query tokens already in RAM, why not compute the pairwise similarities computed if Aij=0? This would still retain some speed?
-	Lines 226-241: what is token at rank k? are these top k retrived tokens?

**Limitations:**

The authors discuss the limitations. Given that the work focusses on the training mechanism, it would have been nice to see trainings with other datasets.

---

> ### Author Rebuttal · Authors · 2023-08-08
>
> Thank you for your detailed review. Please read our clarification for any misunderstanding you might have had while reading the paper.
>
> **A1. The paper overcomplicates a simple concept (Eq 4 can be further simplified from what I understand). Max over j would collapse then.**
>
> First of all, the definition of the alignment matrix $A$ is different in Eq 1 and Eq 4 as described in our paper. Eq 1 shows the baseline sum-of-max operator and how the alignment matrix $A$ can be used to describe its algorithm. Here $A_{ij} = 1_{j=\text{argmax}(P_{ij^\prime})}$ where argmax is over $1\leq j^\prime \leq m$ (i.e., single document) and $P_{ij}=q_i^\top d_j$, meaning that $A_{ij}$ is 1 when $P_{ij}$ is the maximum value among $P_{ij^\prime}$ otherwise 0. Hence, we can replace max over $j$ with $A_{ij}$ in Eq 1. On the other hand, Eq 4 shows the approximated sum-of-max for XTR. In XTR, $A_{ij} = 1_{j \in \text{top-k}(P_{ij^\prime})}$ where the top-k is over $1 \leq j^\prime \leq mB$, meaning that $A_{ij}$ will be 1 when $P_{ij}$ is within the top-k retrieved values among all the $P_{ij^\prime}$ within a mini-batch of $B$ documents. The top-k operator here returns the top-k indices (which will be $k_\text{train}$ during training and $k^\prime$ for inference) among the $j^\prime$s. The main difference with Eq 1 is that the $j^\prime$ spans over all tokens from mini-batch documents in Eq 4, not just from a single document as in Eq 1. As a result, unless all $A_{ij}=1$ (which is unrealistic), Eq 4 will be different from Eq 1 and cannot be collapsed.
>
> We hope that our mathematical formulation can provide proper clarification for your concerns. We will add this clarification in the paper and use a different notation for the alignment matrix of XTR (e.g., $A^\text{XTR}$) to prevent any confusion.
>
> **A2. it would be interesting to see how this method performs with training on NQ, TriviaQA, etc.**
>
> It is a common practice to use MS-MARCO for training retrieval models in many recent works. Indeed, Ni et al., 2021 showed that using MS-MARCO transfers better than using NQ when evaluated on the BEIR benchmarks. Most of the recent papers on neural retrieval were all trained on MS-MARCO, such as ColBERT, ColBERT v2, SPLADE v2, ColBERTer, PLAID, etc.
>
> **A3. For the choice of m_i, Eq. 4 would have collapsed to the form of Eq.1 with extra Ai. Why show it in this form and complicate? Isn't Max over j the same as Aij(qi.Tdj)??**
>
> In our paper (176-177), we discuss that Eq 4 would degenerate to Eq 1 when every $A_{ij}$ of Eq 4 is $1$, but this is an unrealistic special case. Based on our description of $A_{ij}$, Eq 4 does not collapse to Eq 1. Please see our response A1 on the description of alignment matrices, which we believe would resolve some of the reviewer’s concerns.
>
> **A4. Line 120. How is f_colbert(Q,D−)=0.2 when qiTdj−>0.8. Failure case very handwavy. Please explain with proper numbers and reasoning. Is Fig2 different tokens for any +ve/-ve documents or just the retrieved tokens?**
>
> it is possible to have $f_\text{colbert}(Q,D−)=0.2$ when $q_i^\top d_j \rightarrow 0.8$ because the document-level score (e.g. $f_\text{colbert}$) is an average score of token-level pairwise scores (i.e., $q_i^\top d_j$). For instance, imagine there are 6 tokens with $q_i^\top d_j=0.1$ and one token with $q_i^\top d_j=0.8$. $f_\text{colbert}(Q,D-)$ would become (0.8 + 0.1*6)/7=0.2.
> The failure case example is presented for the sake of intuitive understanding of the motivation.  For a more formal understanding of the failure case, we also provided a theoretical analysis of the sum-of-max operator in Appendix A. We will bring some of these results into the main text to make it less confusing.
>
> **A5. Lines 134-137. Very handwavy again. Authors should use proper math format to denote in-batch tokens. It is very difficult to understand when Aij=1.**
>
> We believe that our description in A1 would resolve this concern.
>
> **A6. Lines 134-137. Define top-k_train**
>
> $k_\text{train}$ is a hyperparameter that decides how many in-batch tokens should be retrieved. We will clarify this in the paper.
>
> **A7. For the document tokens and query tokens already in RAM, why not compute the pairwise similarities computed if Aij=0? This would still retain some speed?**
>
> If the token vectors are already loaded on RAM, computing all $A_{ij}$ can be done pretty fast  But we cannot simply assume token vectors are already in RAM. In fact, loading/storing the many token vectors in RAM has been one of the most expensive parts of multi-vector retrieval, and how to make it cheaper is the main problem studied by XTR as well as many prior works (ColBERT v2, PLAID, etc).
> The main advantage of XTR is to completely remove the need of storing any token vectors on RAM and score each document solely based on the retrieved tokens and their scores. While this can significantly reduce the RAM requirement for ColBERT scoring, our general response (G1) on the actual latency also shows how much time we can save with RAM vs Disk-based similarity computation as well as with the XTR-style scoring.
>
> **A8. Lines 226-241: what is token at rank k? are these top k retrived tokens?**
>
> Yes, the token at rank k means the k-th retrieved token. We will clarify this in the paper.

---

> > ### Comment · Reviewer_HcB5 · 2023-08-15
> >
> > Most Answers are convincing.
> > At a very high level, you are just using the document tokens retrieved in the first retrieval stage and interpolating document retrieval scores.
> >
> > - By the same logic from lines 115-132, your inference mechanism in Eq. 4 would weigh any D^- much higher than it actually is. This may cause problems. Reason I was asking for training results on other datasets is that this training/inference mechanism you propose might be very specific to MS-MACRO.  Agreed it transfers to BIER zero shot better but I'd expect for a training mechanism to work with other datasets. What are your thoughts on this?
> >
> >
> > - Can you do this additional experiment of doing inference with Eq 4 on Colbert(i guess you have trained colbert model checkpoints)? I want to better understand the effectiveness of your training mechanism. Take your time on this. Will make sure to check back frequently.
> >
> > Will consider increasing score after your response to this.

---

> > > ### Author Response · Authors · 2023-08-19
> > >
> > > Thank you for reading and considering our responses!
> > >
> > > ### Q1: Is XTR's training mechanism specific to MS MARCO?
> > >
> > > Good question. We did an additional experiment training XTR and T5-ColBERT on Natural Questions (NQ). We train with hard negatives from [1]; training and inference used the same hyperparameters as what we originally used for MS MARCO.
> > >
> > > Model | Training data | BEIR NQ NDCG@10
> > > ---|---|---
> > > DPR (number from [2])	| NQ  | 0.474
> > > T5-ColBERT-base 		| MS MARCO | 0.52
> > > XTR-base  			| MS MARCO | 0.53
> > > T5-ColBERT-base  		| NQ| 0.27
> > > **XTR-base**  			| **NQ** | **0.56**
> > >
> > > For T5-ColBERT, we are not able to achieve good performance when training it on NQ. XTR, on the other hand, achieves strong results. It suggest our training mechanism is not specific to MS MARCO.
> > >
> > >
> > > [1] Ni, Jianmo, et al. "Large Dual Encoders Are Generalizable Retrievers." Proceedings of the 2022 Conference on Empirical Methods in Natural Language Processing. 2022.
> > >
> > > [2] Thakur, Nandan, et al. "BEIR: A Heterogeneous Benchmark for Zero-shot Evaluation of Information Retrieval Models."
> > >
> > > ### Q2: Experiment of doing XTR inference (Eq 4, f_xtr’) on ColBERT
> > >
> > > This is indeed an interesting ablation. We reported this experiment in Table 5 of our paper; below is a summary. We trained T5-ColBERT which is ColBERT with T5 as the backbone, then ran inference with XTR’s scoring function (f_xtr’ in Eq. 4).
> > >
> > >
> > > Model 			| inference		 |  MS MARCO MRR@10 | MS MARCO Recall@1k
> > > --- | --- | --- | ---
> > > T5-ColBERT-base	| sum-of-max  (*computationally expensive*)	| 38.8			| 97.8
> > > T5-ColBERT-base	| f_xtr’, no imputation	| 0.0			| 0.0
> > > T5-ColBERT-base	| f_xtr’, top-k’ score	| 27.7			| 91.8
> > > XTR-base		| f_xtr’, no imputation	| 22.6			| 88.7
> > > XTR-base		| f_xtr’, top-k’ score	| 37.4			| 98.0
> > >
> > >
> > > T5-ColBERT relies heavily on the sum-of-max scoring, which requires loading and scoring the full document representation. When scoring with the retrieved token (f_xtr’, no imputation) instead of full document representation, T5-ColBERT's accuracy is very low. Our top-k’ imputation improves T5-ColBERT's accuracy significantly, but it is still not as good as XTR.
> > >
> > > This result, along with the qualitative analysis in Table F.1, suggests that ColBERT's training recipe does not guarantee good token retrieval, thus reranking with full document representation is necessary.
> > >
> > > On the other hand, XTR is trained for token retrieval, so scoring with retrieved tokens alone(f_xtr’) can already give reasonable results. Adding the top-k’ score imputation, XTR can be as accurate as T5-ColBERT without the need to look up and score the full document representation.

---

> > > > ### Comment · Reviewer_HcB5 · 2023-08-21
> > > >
> > > > Thank for the clarifications. Increased score.

---

### Official Review · Reviewer_5SZT · 2023-07-07

**Soundness:** 3 good
**Presentation:** 2 fair
**Contribution:** 4 excellent
**Rating:** 8
**Confidence:** 3

**Summary:**

This paper deals with the problem statement of document retrieval. First, the paper contrasts and explains the differences between single vector and multi-vector retrieval models. As multi-vectors retrieval models perform better due to their accessibility to more tokens, it involves significant inference costs. The authors propose a new model Contextualized Token Retriever XTR which aims to overcome the disadvantages during inference and bring closer to single vector models. Experimental results on BEIR and LoTTE benchmarks show that XTR achieves State-of-the-art results.


Post rebuttal:
I have read the author's response and the rebuttal sufficiently addressed my concerns.

**Strengths:**

Originality:
The paper brought a lot of theoretical analysis showing why and where multi-vector retrieval models fail. This analysis provided a strong justification to their propose approach. In addition, the authors were able to show the effectiveness of XTR on multiple benchmarks.

Significance:
Given the gap that exists between single and multi-vector retrieval models in terms of performance vs computational efficiency, this paper bridges the gap by not only making the retrieval efficient but also improving the performance. Hence, I feel this paper is quite significant.

**Weaknesses:**

Clarity: I feel this paper needs little fine-tuning in clarity. In the introduction, I couldn't understand the problem statement i.e document retrieval they are dealing with but there was a lot of motivation. I believe adding few sentences about the problem statement makes introduction better. Similarly in the experiments section, there could be separate sub-sections for the datasets, model comparisons and experimental settings. It will makes things more coherent.

**Questions:**

None

**Limitations:**

There is sufficient discussion about the limitations after the conclusion section.

---

> ### Author Rebuttal · Authors · 2023-08-08
>
> Thank you for your review.
>
> **A1. Clarity: I feel this paper needs little fine-tuning in clarity.**
>
> We will definitely add a few more sentences to make the problem statement and the experiment sections clearer. For instance, the third paragraph of the introduction has the problem statement where we can add a few sentences for better understanding. Additionally, for a better description of our method, we provided more formal definitions of the alignment matrices in the response A1 to R4.

---

### Official Review · Reviewer_BRdC · 2023-07-07

**Soundness:** 3 good
**Presentation:** 3 good
**Contribution:** 2 fair
**Rating:** 6
**Confidence:** 5

**Summary:**

The XTR model extends the ColBERT neural IR model by removing one of the efficiency issues: candidate documents (selected using a dense vector index, e.g. FAISS) have to be re-scored by loading as much vectors as there are tokens in the document.

The proposed approach simply reduces the number of vectors representing a document. The candidate score is then (supposedly) equal to the final score (since all the vectors are in the FAISS index). The authors show that in practice (on a variety of datasets, include LoTTE, MIRACL and BEIR) performs roughly similarly to ColBERT

**Strengths:**

The paper presents a simple method that improves over ColBERT by reducing the number of vectors per document. The simple strategy used in the paper is a good alternative to harder to implement pipelines like e.g. PLAID or ColBERTer.  The model performs a little bit worse than ColBERT but improves (in theory) the latency, although the latter is not measured in the paper.

The experimental work is well conducted on a variety of dataset, showing the robustness of the proposed approach.

**Weaknesses:**

- No experiment measuring the observed latency are reported. While the estimated FLOPs/query is important, seing actual difference on the same hardware would strengthen the message

- No comparison with PLAID which implements a lot of strategies to improve the efficiency of ColBERT (which is another way to deal with the problem of the number of vectors in documents). More importantly, another very related work, ColBERTer (CIKM 2022) which also reduces the number of vectors per document using a sparsity-inducing loss, is missing.

- Figure 4 should report the recall of gold tokens (rather than the precision)

**Questions:**

- Adding a sparsity loss would have been an alternative to the learning scheme: why was this not used since it might give more stability to the process?

- What is the real latency time of such a model when the number of top-K vectors increases

- is $k_{train}$ the same thing as $m$ in eq. 2?

**Limitations:**

The limitations section is a bit too generic (not really related to the proposed approach). See weaknesses for possible issues to report at this level.

---

> ### Author Rebuttal · Authors · 2023-08-08
>
> Thank you for your review.
>
> **A1. No experiment measuring the observed latency are reported.**
>
> Since it is difficult to reimplement every baseline within our hardware and infrastructure, apples-to-apples latency comparisons are a bit tricky. Admitting the differences in implementations, we report the latency of XTR in our general response G1. To summarize, our implementation of XTR runs reasonably fast while significantly improving the speed of T5-ColBERT, both of which are tested in the same environment.
>
> **A2. No comparison with PLAID or ColBERTer**
>
> PLAID and ColBERTer optimize the first-stage token retrieval, which is orthogonal to the contribution of XTR (better scoring stage without the gathering stage) and they can be applied to XTR as well. For instance, sparse optimization techniques of AligneR (Qian et al., 2022), which can be more directly compared to ColBERTer as it optimizes the first-stage token retrieval, can be applied to XTR without minimum performance losses. Please also refer to our general response G2 for the detailed comparisons.
>
> **A3. Figure 4 should report the recall of gold tokens (rather than the precision)**
>
> While Figure 4 reported the precision of token at the k-th rank, we agree that the recall of gold tokens would be interesting to see since it would demonstrate how well the token retrieval serves as the first stage of the multi-vector retrieval. We found that indeed, the token retrieval of XTR has higher token recall (i.e., how many tokens of all gold documents are retrieved?), achieving 0.2517 recall@1000 while T5-ColBERT achieves 0.1047 recall@1000 (TREC-COVID). Note that the overall recall would be very low since the model doesn’t have to retrieve the entire set of gold tokens, which include many stopwords as well. Similar results were observed on MS-MARCO and other BEIR benchmarks, which translate to the superior document-level recall@100 of XTR (Table E.1 in Appendix). We also added figures of the token retrieval recall in the pdf attached in the general response, which we will include in our final draft.
>
>
> **A4. Adding a sparsity loss would have been an alternative to the learning scheme: why was this not used since it might give more stability to the process?**
>
> Exploring the sparsity loss was extensively studied in the prior work. In particular, AligneR (Qian et al., 2022) tested the Optimal Transport and L1 regularization techniques to sparsify the pairwise as well as unary alignments. However, while these sparsification techniques make the model decide which tokens of a document are salient, we found that existing sparsification methods do not encourage those salient tokens to be retrieved across other documents. This is primarily due to the fact that sparsification methods work independent of the query representations.
>
> **A5. What is the real latency time of such a model when the number of top-K vectors increases**
>
> For the real latency of each stage (retrieval and scoring), please refer to the general response G1 where we show some fair comparisons with increasing top-k vectors.
>
> **A6. is k_train the same thing as m in eq. 2?**
>
> k_train is a hyperparameter that decides which top tokens in a mini-batch should be taken into considerations for the scoring of mini-batch documents. On the other hand, m is the length of a document $D$ (i.e., number of tokens in $D$).

---

> > ### Comment · Reviewer_BRdC · 2023-08-14
> >
> > Thanks for your answers. I appreciate the effort with respect to latency results, I think they are important.
> >
> > I still disagree on the positioning of the paper with respect to ColBERTer or PLAID – XTR also tries to increase efficiency at the cost of effectiveness, although the approach is different. I agree however that some methods could be complementary (although it is far from clear if these would lead to additive improvements).

---

> > > ### Author Response · Authors · 2023-08-15
> > >
> > > Thank you for the discussion! To show that the contributions of XTR can be complementary with one of the baselines, we implemented a sparsity-inducing loss following [2] to prune unimportant token vectors from the document, which is similar to the pruning done in ColBERTer. The results are as follows:
> > >
> > > Model | % document vectors reduced | MS MARCO MRR@10 | MS MARCO Recall@1k
> > > ---|---|---|---|
> > > XTR-base 					| 0%  	| 0.377 | 0.981
> > > XTR-base, with pruning 			| 29.2%| 0.371 | 0.980
> > > ColBERTer, without pruning* 	| 0% 	| 0.387 | 0.960
> > > ColBERTer, with pruning* 	| 29% 	| 0.387 | 0.961
> > >
> > > *: ColBERTer results are from Table 3 reported in ColBERTer paper [1]. ColBERTer's MRR@10 is slightly higher than XTR-base, as ColBERTer uses distillation in training, an extra CLS token for document representation, etc.
> > >
> > > Our results show that one could further reduce 30% document token vectors with ColBERTer’s vector pruning technique, while still using XTR to simplify and speedup the refinement stage, without hurting the quality. This shows how the improvements can be additive.
> > >
> > > For PLAID, more investigation would be needed since it requires significant changes on the first stage token retrieval, which we leave as future work.
> > >
> > > [1] Hofstätter, Sebastian, Omar Khattab, Sophia Althammer, Mete Sertkan, and Allan Hanbury. "Introducing neural bag of whole-words with ColBETer: Contextualized late interactions using enhanced reduction." In Proceedings of the 31st ACM International Conference on Information & Knowledge Management, pp. 737-747. 2022.
> > >
> > > [2] Qian, Yujie, Jinhyuk Lee, Sai Meher Karthik Duddu, Zhuyun Dai, Siddhartha Brahma, Iftekhar Naim, Tao Lei, and Vincent Y. Zhao. "Multi-vector retrieval as sparse alignment." arXiv preprint arXiv:2211.01267 (2022).

---

### Official Review · Reviewer_VrQp · 2023-07-10

**Soundness:** 3 good
**Presentation:** 3 good
**Contribution:** 3 good
**Rating:** 6
**Confidence:** 3

**Summary:**

This work proposes XTR, ConteXtualized Token Retriever, which is a method for multi-vector retrieval with a simple and effective objective function. Comparing to prior work on multi-vector retrieval method ColBERT where all the tokens from query and candidate document needs to be computed in order to calculate the final query/document score, XTR only uses retrieved tokens in the document to calculate the score, which greatly reduces inference time. It also provided analysis on the situations when ColBERT training objective may fail and proposed new objective that can mitigate the issue.
In experimental session, extensive experiments haven been done to show the effectiveness of the proposed method. XTR achieves competitive performances with current SOTA multi-vector retrieval models, while being much efficient at inference time. It also achieved SOTA in zero-shot retrieval benchmark BEIR and strong results in multi-lingual retrieval benchmarks.
Finally, analysis was presented in section 5 to shed lights on why XTR presents a better token retrieval.


**Strengths:**

- It proposes a method for multi-vector retrieval which achieved strong performances on multiple benchmarks and can greatly reduce inference time comparing to prior work on multi-vector retrieval
- The proposed method achieved SOTA results on zero-shot retrieval benchmarks and multilingual retrieval benchmarks
- It provides insightful analysis on where previous multi-vector retrieval method (ColBERT) failed and proposed new training schema that solve the problem. It also provides examples that validate the proposed methods in qualitative analysis.
- It is well written overall with thoughtful analysis and extensive experimentation results

**Weaknesses:**

- The XTR method is mainly built on T5 models, without exploration on other architectures.
- The analysis on equation (4) can be more elaborate (see Q1).

**Questions:**

Q1: Could you provide more insights on why the upperbound of m_i in equation (4) works? Have you tried other bounds/estimations?

Q2: In table 1, how does the estimated FLOPs/query looks like for single vector retrieval methods such as DPR?

**Limitations:**

- The authors listed that XTR is trained on MSMARCO dataset which may have license issue.
- The proposed method only trained on T5 models, and no other type of architecture has been explored.

---

> ### Author Rebuttal · Authors · 2023-08-08
>
> Thank you for your detailed review on our work.
>
> **A1. The XTR method is mainly built on T5 models, without exploration on other architectures.**
>
> We mainly used T5 models (encoder-only) since it is easier to scale the architecture (from base to xxl) and has been shown to work well for the initialization of dual encoders such as GTR and Sentence-T5. In particular, GTR and Sentence-T5 have already been shown to work equal or better than BERT-based encoders such as DPR or Sentence-BERT. In the future, we are planning to use stronger and larger pre-trained language models such as PaLM or LLAMA, but using decoder-only LMs as retrievers is out-of-scope in this work.
>
> **A2. The analysis on equation (4) can be more elaborate (also related to Q1).**
>
> Please see our response A1 to R4 on a more formal description of the alignment matrices. Specifically, we defined the alignment matrix in a more mathematical way giving a clear distinction between Eq 1 and Eq 4.
>
> **A3. Could you provide more insights on why the upperbound of m_i in equation (4) works? Have you tried other bounds/estimations?**
>
> The success of our upper bound-based approximation tells us that estimating missing value in a query-dependent manner is more important than exactly imputing missing value. As reported in Table 5, constant imputation does not work well. On the other hand, more sophisticated imputation methods such as power-law based imputation, which was omitted from the paper for brevity, did not provide significant improvement despite its complexity. As a result, we use the upper bound-based estimation, which is simple to utilize while robust enough to provide good estimation. We will report the power-law based imputation method in the final version.
>
> **A4. In table 1, how does the estimated FLOPs/query looks like for single vector retrieval methods such as DPR?**
>
> For a single vector retrieval, FLOPs/query is much lower since it does not have multiple representations of each query and document. Note that single-vector retrieval does not have the scoring stage since the retrieval directly provides final scores and rankings. Assuming sublinear MIPS, dual encoders like DPR/GTR would have $2d \log L$ FLOPs/query for retrieval where $L$ is the number of documents (e.g. $d=768, L=5\times10^6$ for DPR). On the other hand, naive multi-vector retrieval models would have $2nd \log M$ where n is the number of query tokens and $M$ is the number of document tokens in a corpus (e.g. $d=128, n=16, M=1\times 10^9$ for XTR). While our focus in this work is to better optimize the scoring stage in terms of FLOPs/query and the memory usage, previous works such as PLAID or ColBERTer are worth mentioning, which reduce the complexity of the first-stage token retrieval as described in our general response G2.

---

### Author Rebuttal · Authors · 2023-08-08

We thank all the reviewers for their thoughtful comments and feedback.

Most of our reviewers agree that XTR effectively mitigates the problem of three-stage inference of multi-vector models and provides significant improvements over strong baseline models. Some of the main concerns include latency reports (with comparison against the existing baselines) (R2, R5) and the presentation quality of the paper (R1, R3, R4).

**G1. Actual Latency of XTR**

Our initial manuscript did not compare the actual latency due to the differences in libraries and infrastructure used for implementing each baseline and XTR. Specifically, XTR uses ScaNN (Guo et al., 2020), which has different optimization techniques and a different distributed system compared to Faiss (Johnson et al., 2019) used by various baselines. XTR also does not use any centroid-based approximation used by ColBERTv2 or PLAID. Nevertheless, as many reviewers (R2, R5) asked for, the actual latency might provide a clearer sense of the contribution of XTR. Under the same environment, we report the latency of the token retrieval as well as the scoring stage, which uses naive CPU-based Numpy without any multi-processing. The token retrieval stage uses CPU-based distributed ScaNN (same for T5-ColBERT and XTR). For T5-ColBERT, document token vectors are loaded for the exact sum-of-max, either from Disk or RAM. MS-MARCO dev set queries were used to average per-query latencies.

| Top-k’ tokens | Token retrieval | T5-ColBERT-Disk scoring | T5-ColBERT-RAM scoring | XTR-scoring | XTR total latency |
| --- | :---: | :---: | :---: | :---: | :---: |
| 1,000 | 0.25 sec | 11.56 sec | 3.13 sec | 0.38 sec | 0.63 sec |
| 4,000 | 0.31 sec | 47.05 sec | 12.67 sec | 1.64 sec | 1.95 sec |
| 40,000 | 0.47 sec | 8 min 2.09 sec | 2 min 7 sec | 17.17 sec | 17.64 sec |

Note that these results are mainly for the fair comparisons against the full sum-of-max baseline (i.e., T5-ColBERT) so we did not use multi-processing. Our implementation of the XTR scoring with multiprocessing actually takes about < 0.3 sec even when $k^\prime=40,000$, hence giving < 0.8 sec in total. The scoring stage of XTR also does not require loading $\mathcal{O}(nk^\prime\bar{m}d )$ floating points from Disk or RAM as T5-ColBERT does, which can range from 450MB ($k^\prime=1,000$) to 18GB ($k^\prime=40,000$) per query.

**G2. Comparison against PLAID and ColBERTer**

Prior work such as PLAID and ColBERTer focus on different aspects of multi-vector retrieval. PLAID and ColBERTer focus on improving the efficiency of the first stage token retrieval, while XTR aims to greatly simplify the gathering and scoring stages as well as improve the training objective.  More specifically, PLAID 1) uses cluster-centroid based approximated search to provide a small number of candidates and 2) looks up the candidate vectors to apply the full sum-of-max over them. ColBERTer proposes to use whole-word representations with more aggressive dimensionality reduction, reducing the cost of first-stage retrieval. XTR, on the other hand, 1) uses a naive MIPS library that provides a large number of candidates and 2) uses the light-weight scoring function, which eliminates index lookup. XTR additionally modifies the training objective offering better token retrieval. We believe that these paradigms are very orthogonal and hope to explore combining these efforts in the future. For instance, it would be interesting to see if the training objective of XTR could improve the candidate generation part of PLAID so that it can further have a smaller number of candidates.

**G3. Presentation Quality**

While many reviewers (R1, R2, R5) believe that the presentation of our paper is either good or excellent, we acknowledge that there is room for improvement as R3 and R4 suggested. In particular, we provided detailed clarification on each concern of R4 regarding the equations and examples (please see R4-A1), which will be added in our manuscript. For a better introduction, we will add clarifying sentences as R3 suggested (please see R3-A1).

We also attached figures related to the question raised by R2-A3.

---

### Decision · Program_Chairs · 2023-09-21

**Decision:**

Accept (poster)

**Comment:**

The paper is a clear accept. It identifies a practical problem and provides a principled solution, based on a neat generalization of the multi-vector retrieval objective, with strong empirical results. The reviews are positive and the author response adequately addressed the reviewers' questions.